# Evaluation of the Dynamic Core of the PALM Model System 6.0 in a Neutrally Stratified Urban Environment: Comparison between LES and Wind-tunnel Experiments

Tobias Gronemeier[1], Kerstin Surm[2], Frank Harms[2], Bernd Leitl[2], Björn Maronga[1,3], and Siegfried Raasch[1]

[1]Leibniz University Hannover, Institute of Meteorology and Climatology, Hannover, Germany
[2]Universität Hamburg, Meteorological Institute, Hamburg, Germany
[3]University of Bergen, Geophysical Institute, Bergen, Norway

**Correspondence:** Tobias Gronemeier (gronemeier@muk.uni-hannover.de)

**Abstract.** We demonstrate the capability of PALM 6.0, the latest version of the PALM model system, to simulate neutrally stratified urban boundary layers. The studied scenario includes a real-case building setup of the HafenCity area in Hamburg, Germany. Simulation results are evaluated against wind-tunnel measurements of the same building layout utilizing PALM's virtual measurement module. The comparison reveals an overall very high agreement between simulation results and wind-tunnel measurements not only for mean wind speed and direction but also for turbulence statistics. However, differences between measurements and simulation arise within close vicinity of surfaces where the resolution prevents good representation of the building layout. In the end, we discuss how these differences can be reduced using already implemented features of PALM.

## 1 Introduction

The PALM model system version 6.0 is the latest version of the computational fluid-dynamics (CFD) model PALM, a Fortran based code to simulate atmospheric and oceanic boundary layers. Version 6.0 was developed within the scope of the Urban Climate Under Change ([UC]²) framework funded by the German Federal Ministry of Education and Research (Scherer et al., 2019; Maronga et al., 2019). The aim of the [UC]² project is to develop a fully functional urban climate model capable of simulating the urban canopy layer from city scale down to building scale with grid sizes down to $1\,\mathrm{m}$. A detailed description of the model system is given by Maronga et al. (2015, 2020a). PALM has already been applied in a variety of studies within the area of urban boundary-layer research (e.g., Letzel et al., 2008; Park et al., 2012; Kanda et al., 2013; Kurppa et al., 2018; Wang and Ng, 2018; Paas et al., 2020). Built upon PALM version 4.0, the latest version 6.0 contains many new features and improvements of already existing components of the model system. One of the most impacting changes is the new treatment of surfaces within PALM. While PALM did not distinguish between different surface types within former versions, it is now possible to directly specify a surface type to each individual solid surface within a model domain via the land-surface model (Maronga et al., 2020a) or the building surface model (Resler et al., 2017; Maronga et al., 2020a). Also, a fully three-dimensional obstacle

representation is possible while former versions allowed only a 2.5-dimensional representation of obstacles (no overhanging structures like bridges or gates). These additions, however, required extensive re-coding of the former version PALM 4.0, which affected also the dynamic core of the model. This also includes the modularization of the code base put further to practice,

which lead to a re-ordering and re-grouping of code parts into several internal modules like a constant flux-layer module, boundary-conditions module or turbulence-closure module. Changes to the dynamic core are, however, limited to technical changes, i.e. the underlying physical equations are still identical to the previous version 4.0.

Whereas former versions of PALM were already evaluated against wind-tunnel measurements, real-world measurements, and other CFD codes (Letzel et al., 2008; Razak et al., 2013; Park et al., 2015; Gronemeier and Sühring, 2019; Paas et al.,

2020), the significant changes of PALM's code base produces different results compared to former versions. These changes are either due to round-off errors purely because some code parts are not executed within the same order as before but may also be due to formerly unknown code defects being fixed or new defects being introduced. Hence, the updated version 6.0 requires a new evaluation from scratch. A sufficient evaluation is inevitable to ensure confidence in the results of the PALM model system as it is also the case for every other CFD code (Blocken, 2015; Oberkampf et al., 2004).

Because of the high complexity of PALM, evaluating the model is a very lengthy and costly exercise and a complete validation of all model components would easily go beyond the scope of any single article. Within this study, we therefore focus to evaluate the model's flow dynamics which make up the core of the model system and build the foundation for all other features within PALM. In order to isolate the pure dynamics from all other code parts, PALM is operated in a pure dynamic mode, i.e. all thermal effects (temperature and humidity distribution, radiation, surface albedo, heat capacity, etc)

are switched off. The simulation results can then be evaluated using wind-tunnel measurements that are recorded in a similar setup as the simulation data as stated by Leitl and Schatzmann (2010). While it is virtually impossible to neglect temperature or humidity effects in real-world measurements, wind-tunnel experiments can provide exactly the same idealized conditions as used in our idealized simulation. Also, other difficulties such as additional non-resolved obstacles like trees or sub-grid features on building walls existing in the real world can make a comparison with real-world measurements troublesome (Paas

et al., 2020). Paas et al. (2020) compared PALM simulations to measurements of a mobile measurement platform. Although overall good agreement was found between PALM and the measurements, some non-resolved obstacles like trees complicated the comparison at several points and led to differences in results. Hence, we decided to compare PALM against an idealized wind-tunnel experiment for this study.

A realistic building setup, in this case the HafenCity area of Hamburg, Germany, is chosen for the study. A real-case building

setup has the advantage over more idealized, e.g. a single-cube, cases that a variety of different building configurations, also including more or less solitary buildings, can be covered in a single simulation. A real-case building setup has the advantage to include a variety of building configurations ranging from solitary buildings to complex street canyons within a single simulation. Likewise, it may show the capability of PALM to correctly reproduce a complex realistic wind distribution.

The evaluation study was originally designed as a blind test where only the boundary conditions (building layout, approach-

ing flow profile, location of measurements) but no further results of the wind-tunnel experiment was available to conduct the PALM simulation. Such a blind test has the benefit to prevent model tuning and shows how well a model can reproduce ref-

erence data only based on the boundary conditions. This also reflects a more realistic use case where reference data might not even exist. However, after comparing results from both PALM and wind-tunnel experiments, several errors within the simulation setup were identified like errors in building height and the roughness representation within the upwind region. The PALM setup was then updated with all identified flaws corrected and the case was re-simulated. Even though there are methods to adjust CFD results to better match to measurements (e.g., Blocken et al., 2007), these adjustments depend on the individual case and need to be re-calculated for each new studied situation and are also only available if detailed reference data are available. Such setup tuning was not considered for the update of the simulation setup. Corrections solely considered information that was already available during the first simulation but was just not considered (layout of roughness elements within the wind tunnel) or simply incorrect (wrong building heights).

## 2 Experimental setup

### 2.1 Wind-tunnel experiment

Measurements were carried out at the Environmental Wind Tunnel Laboratory (EWTL) facility 'WOTAN' at the University of Hamburg, Germany. The $25\,\mathrm{m}$ long wind tunnel provides an $18\,\mathrm{m}$ long test section equipped with two turn tables and an adjustable ceiling. The cross section of the tunnel measures $4\,\mathrm{m}$ in width and $3\,\mathrm{m}$ in height. Figure 1 shows a photograph from within the wind tunnel for reference. For each wind tunnel campaign, a neutrally stratified boundary layer flow is generated by a carefully optimized combination of turbulence generators at the inlet of the test section, and a compatible floor roughness. For the present study a boundary layer flow was modelled to match full scale conditions for a typical urban boundary layer measured at a $280\,\mathrm{m}$ tall tower in Billwerder, Hamburg. The mean wind profile can be described by a logarithmic wind profile with a roughness length $z_0 = (0.66 \pm 0.22)\,\mathrm{m}$ and by a power law with a profile exponent $\alpha = 0.21 \pm 0.02$. The approaching flow profile is depicted in Fig. 2 and was modelled at a wind direction of $110°$.

An area of $2.6\,\mathrm{km}^2$ covering the HafenCity of Hamburg, Germany, was modelled at a scale of $m = 1/500$ within the wind tunnel (see Fig. 1). Scaling of space $l$, time $t$ and velocity $u$ between model scale (ms) and full scale (fs) is then achieved via

$$l_{\mathrm{fs}} = \frac{l_{\mathrm{ms}}}{m}, \tag{1}$$

$$t_{\mathrm{fs}} = \frac{t_{\mathrm{ms}}}{m}, \tag{2}$$

$$u_{\mathrm{fs}} = u_{\mathrm{ms}}. \tag{3}$$

A 2D Laser-Doppler-Anemometry (LDA) System was used to measure component-resolved flow data at sampling rates of $200\,\mathrm{Hz} - 800\,\mathrm{Hz}$ (model scale), resolving even small-scale turbulence in time at most but unfortunately not all measurement locations. At each measurement location a 3 minutes time series was recorded, which corresponds to a period of about 25 hours at full scale. The reference wind speed was permanently monitored close to the tunnel inlet through Prandtl tube measurements. For the model evaluation case presented here, measurements were taken at 25 different locations within the building setup as shown in Fig. 3. As the measurements were originally planned and used for a different study focusing on near-ground ventila-

tion and pedestrian wind comfort, locations were not specifically chosen for the present study. However, the measurements still cover different aspects of the flow within the building canopy including open areas, narrow and wide street canyons as well as intersections.

## 2.2 PALM simulation

The PALM Model System 6.0, revision 3921, was used to conduct the simulation for this study. PALM was operated using a fifth-order advection scheme after Wicker and Skamarock (2002) in combination with a third-order Runge-Kutta time-stepping scheme after Williamson (1980). At this point, we skip a detailed description of the PALM model, which is provided by Maronga et al. (2015, 2020a). The simulation was conducted at full scale with a domain covering an area of $6000\,\mathrm{m}$ by $2880\,\mathrm{m}$ horizontally and $601\,\mathrm{m}$ vertically at a spacial resolution of $\Delta x = \Delta y = \Delta z = 1\,\mathrm{m}$ in each direction. This resulted in about $10.4 \times 10^9$ grid points for the used staggered Arakawa C grid (Harlow and Welch, 1965; Arakawa and Lamb, 1977). The area of interest, i.e. the HafenCity area, was situated downstream of the simulation domain. The model domain was oriented so that the mean flow direction was aligned with the $x$-direction. With a mean wind direction of $110°$, the model domain was hence rotated counter-clockwise by $200°$.

The building layout as used in PALM is depicted in Fig. 4. In PALM, topography is considered using the mask method (Briscolini and Santangelo, 1989) where a grid volume is either $100\,\%$ fluid or $100\,\%$ obstacle. In combination with PALM's rectilinear grid, this can cause buildings not aligned with the grid to appear differently, more brick-like, than they were within the wind tunnel.

The basic setup for this study is based on the settings used in the former study of Letzel et al. (2012).

A heterogeneous building setup usually requires a non-cyclic boundary condition along the mean flow direction to ensure that building-induced turbulence is not cycled over and over the analysis area which otherwise might influence the results. However, tests with non-cyclic boundary conditions along the mean flow showed that extremely long simulation times would be required to generate a stationary state. Hence, we used cyclic boundary conditions instead, which reduced the required CPU-time significantly. The domain was extended in mean flow direction ($x$-direction) to allow the building-induced turbulence to dissipate before the flow hits the target area again due to the cyclic conditions. As the simulation was aimed at a pure neutral case and without releasing any trace gases or alike within the city area, there was no disadvantage in using cyclic boundary conditions instead. After a simulation time of 1.5 hours steady-state conditions were reached.

A constant flux layer was assumed between the surface and the first computational grid level to calculate the surface shear stress. The exact value of the roughness length $z_0$ for the building surfaces is not known from the wind-tunnel experiment. Therefore, it was estimated as $z_0 = 0.01\,\mathrm{m}$ which also satisfied the general recommendation by Basu and Lacser (2017) who state $z_0 \leq 0.02 \cdot \min(\Delta z)$. Due to the staggered grid, the first computational level was positioned $0.5\Delta z$ above the surface, hence, $z_0 = 0.02 \cdot 0.5 \cdot 1\,\mathrm{m} = 0.01\,\mathrm{m}$.

For the approaching flow, the modelled roughness length was estimated as $z_0 = (0.66 \pm 0.22)\,\mathrm{m}$ within the wind-tunnel experiment (cf. Sect. 2.1). With the chosen resolution of $1\,\mathrm{m}$ such large roughness cannot be represented by be used surface-flux parameterization, but needs to be explicitly simulated by resolved-scale roughness elements. They were placed within the

simulation domain of the exact same shape and layout as they were present in the wind-tunnel experiment. This produced a similar boundary layer flow within both experiments as shown in Fig. 2.

To match the conditions within the wind tunnel, a strictly neutral atmosphere was considered with potential temperature being constant over time. Also, the Coriolis force was neglected.

In the past, persistent streak-like structures oriented along the mean-wind direction were reported for LES of neutral flows using cyclic boundary conditions (Munters et al., 2016). Such streaks naturally develop within the neutral boundary layer and reach lengths of several kilometres and move along the mean wind direction while not moving in span-wise direction. They form randomly and have a limited lifetime. In combination with cyclic boundary conditions, however, the start and end of a streak can merge forming an infinite streak that is self-containing and persistent in time. To avoid the artificial persistence of these structures by cyclic boundary conditions, a shifting method was used according to Munters et al. (2016). This method breaks up the infinite and persistent streak-like structures and ensures a natural dissipation. The flow was shifted by $300\,\mathrm{m}$ in $y$-direction, i.e. perpendicular to the mean wind direction, before entering the domain at the left boundary.

The wind field was initialized using a turbulent wind field from a precursor simulation via the cyclic-fill method (Maronga et al., 2015). The setup of the precursor simulation was similar to the main simulation but with a reduced domain size of $600\,\mathrm{m}$ by $600\,\mathrm{m}$ in horizontal direction. To initialize the precursor simulation, the normalized approaching wind profile as measured in the wind tunnel was used and scaled to a wind speed of $4\,\mathrm{m\,s^{-1}}$ at $50\,\mathrm{m}$ height to get a representative wind speed for within the canopy layer. This resulted in a fixed wind speed of $6.26\,\mathrm{m\,s^{-1}}$ at the top boundary for the precursor and main simulation.

The total simulation time of the main simulation was 4 hours of which the first 1.5 hours were required to reach a steady state of the simulation. The final 2.5 hours were used for the analysis.

Figure 2 shows the mean wind profile of the flow approaching the building area during the analysis time as well as the approaching flow of the wind-tunnel experiment. Note that the street-level height is defined as $z = 0\,\mathrm{m}$ while the lower-most height was at water level which is $5\,\mathrm{m}$ below street level (cf. Fig. 4). Hence, the shown approaching wind profile starts at $z = -5\,\mathrm{m}$.

## 2.3 Measurement stations

Within the wind-tunnel experiment, wind speed was measured at certain measurement stations within the building array. The locations of which are shown in Fig. 3. To be able to mirror the measurements as best as possible, the virtual measurement module of PALM was used (Maronga et al., 2020a). This module allows to define several virtual measurement stations within the model domain via geographical coordinates. The model domain itself then needs to be geo-referenced in order to identify the grid points closest to the measurement location. Referencing is done by assigning geographical coordinates and orientation to the lower left corner of the domain.

When mapping the measurement stations onto the PALM grid, there were two difficulties: First, there was not always a grid point available at the exact location of the measurement within the wind-tunnel experiment. Therefore, measurement positions are slightly shifted between both experiments by a distance less than $1\,\mathrm{m}$. Second, the topography in close vicinity of a measurement point might have been slightly different due to topography representation used in PALM (see Sect. 2.2). To

overcome these two issues, virtual measurements not only from the closest grid point to a measurement position were saved but also values from the neighbouring grid points. In post-processing, the area of each measurement station was analyzed and a grid point selected that best fitted the wind-tunnel measurements.

At each measurement station presented in this study, vertical profiles were recorded with a sampling rate between $8.7\,\mathrm{Hz}-11.2\,\mathrm{Hz}$ (measurements recorded during each time step).

## 3 Results

### 3.1 PALM simulation

The PALM simulation required a spin-up time of 1.5 hours as can be seen by the time series of the domain-averaged kinetic energy $E = 0.5\sqrt{u^2 + v^2 + w^2}$ and the friction velocity $u_*$ (see Fig. 5). Both quantities stabilized after 1.5 hours at around $E = 15.4\,\mathrm{m^2\,s^{-2}}$ and $u_* = 0.16\,\mathrm{ms^{-1}}$. Therefore, only data from the last 2.5 hours of the simulation were used for the following evaluation.

The horizontally and time-averaged vertical profile of the stream-wise component of the vertical momentum flux $wu$ is shown in Fig. 6. The vertical momentum flux $wu$ can be split into a resolved and a sub-grid scale (SGS) part which is parameterized via an SGS model. The higher the resolved part, the less the SGS model contributes to the flux therefore indicating that the flux and hence the turbulence causing the flux is well resolved. The ratio of the resolved and the total momentum flux (total meaning resolved plus SGS part) is close to 1 revealing that turbulence is properly resolved within the simulation domain (see Fig. 6). At the surface, turbulence is less resolved due to the fact that turbulent structures tend to become smaller the closer they get to the surface and cannot be resolved by the grid spacing any more (Maronga et al., 2020b). However, the ratio between resolved and total $wu$ is above 0.9 except for the lowest two grid levels, where the ratio drops down to 0.78. The small disturbance that is visible at $z = 15\,\mathrm{m}$ is related to the roughness elements. Most of these elements reach up until $z = 15\,\mathrm{m}$ causing the disturbance in the vertical $wu$ profile at that height.

To get an impression of the turbulent structures, Fig. 7 shows a snapshot of the magnitude of the three-dimensional vorticity as a measure of turbulence. One can clearly identify strong turbulent features (yellow and red structures) within the vicinity of buildings while only weak turbulence is present above smooth surfaces. Strong turbulence outside of the building array is caused by roughness elements that are not visible within the figure.

### 3.2 Comparison between wind-tunnel and PALM

To compare both experiments, results must be normalized first as the experiments were conducted on different scales and used different mean wind speeds. The reference wind speed $u_\mathrm{ref}$ used for normalization corresponds to the wind speed of the approaching flow at a height of $50\,\mathrm{m}$ (full scale). The reference height was defined by previously conducted laboratory experiments to be representative for the measured canopy flow and is expected to be well within the height range for that a

scaled neutrally-stratified atmospheric boundary layer wind flow could be modeled most accurately. In the following, results are given at full scale if not stated otherwise.

Figure 8 shows the wind distribution for each measurement station at the lowest measurement height for (a) the wind-tunnel measurements ($z = 3\,\mathrm{m}$) and (b) the PALM simulation ($z = 2.5\,\mathrm{m}$). Due to the staggered grid used in PALM (cf. Sect. 2.2), PALM measurements are positioned $0.5\,\mathrm{m}$ below their corresponding wind-tunnel measurements. Note, that measurement station 15 (cf. Fig. 3) was positioned on top of a building where measurements were only available above $18\,\mathrm{m}$ height. At most measurement stations, the main wind direction is similar in the PALM simulation compared to the wind-tunnel data.

Noticeable differences of the wind distribution occur at stations 6, 7, 10, and 20, where a larger variation in wind direction or a different mean wind direction is reported within the PALM simulation. On average, wind speed is about $9\,\%$ less in the PALM simulation compared to the wind-tunnel measurements.

At the next measurement height (wind tunnel: $10\,\mathrm{m}$; PALM: $9.5\,\mathrm{m}$), the wind distribution is still very close between PALM and the wind tunnel at most stations (see Fig. 9). Stations 6, 10 and 20 still show noticeable differences. The difference in average wind speed reduces to $5\,\%$ between PALM and wind-tunnel results. This difference reduces to less than $2.5\,\%$ at $40\,\mathrm{m}$ height and above.

A more detailed comparison of the difference in wind speed is given by Fig. 10 which shows scatter plots of wind-tunnel and PALM measurements at each station and height which are 173 data pairs in total. Looking at the horizontal wind speed $U_{\mathrm{hor}}$ and the wind-speed components $u$ and $v$, PALM underestimates the lower wind speeds while higher wind speeds compare well to the wind-tunnel measurements. Wind direction $d$ differs by less than $4°$ on average with a maximum difference of less than $44°$. It has to be noted, however, that wind tunnel measurements might be located between grid points of the PALM grid creating a spacial offset between the measurements. Especially close to obstacles, this spacial offset can lead to differences between both experiments.

The general lower wind speed recorded within PALM has three major reasons: (i) a miss-match in measurement height, (ii) a miss-match in $z_0$ between both experiments and (iii) the step-wise building representation caused by PALM's rectilinear grid. Within PALM, measurements were located $0.5\,\mathrm{m}$ lower than in the wind-tunnel experiment due to the staggered Arakawa C grid ($u$ and $v$ values are calculated at half the height of each grid cell; given that $\Delta z = 1\,\mathrm{m}$, $u$ and $v$ are hence calculated at heights of $0.5\,\mathrm{m}$, $1.5\,\mathrm{m}$, $2.5\,\mathrm{m}$, and so on) and were not interpolated to different heights in order to not alter the simulation results by adding additional uncertainty due to the chosen interpolation techniques. When comparing PALM results at $0.5\,\mathrm{m}$ above the wind-tunnel measurements, the underestimation of wind-speed reduces to $5\,\%$ at $3\,\mathrm{m}$ height. Because vertical gradients of the wind-speed decrease with height, differences in measurement heights are less severe at greater heights.

Second, a miss-match of $z_0$ between both experiments also affects results most significantly at the lowest height levels. This is supported by the fact that the largest difference in mean wind speed ($9\,\%$ lower wind speed) is observed at the lowest measurement height. Hence, the surfaces within the wind-tunnel experiment might have been smoother than estimated for the PALM simulation and $z_0 = 0.01\,\mathrm{m}$ might have been too large. However, in a different not yet published wind-tunnel experiment with similar wall materials of the building model, roughness lengths were observed between $0.002\,\mathrm{m}$ and $0.01\,\mathrm{m}$. This puts the chosen $z_0$ for the simulation at the upper end of the possible value range for the roughness within the wind-tunnel experiment.

The third reason affects results within the entire building canopy layer. Because PALM discretizes obstacles on a rectilinear grid as mentioned in Sec. 2.2, smooth building walls are represented by step-wise surfaces if they are not aligned with the grid layout. Therefore, building walls become significantly rougher than they were in the wind-tunnel experiment. This causes higher turbulence and an overall reduced mean wind speed within the building canopy layer.

To better evaluate the deviations between both experiments, different validation metrics were calculated. Within COST Action 732 (Schatzmann et al., 2010), several validation metrics are listed to help evaluating simulation models. The proposed metrics are the factor-of-two $FAC2$, the hit rate $q$, the fractional bias $FB$, the geometric mean bias $MG$, the normalized mean square error $NMSE$, and the geometric variance $VG$. Additionally, also the correlation coefficient $R$ is calculated. These metrics are defined as:

$$FAC2 = \frac{1}{N}\sum_i n_i \ , \ n_i = \begin{cases} 1 & \text{if } \frac{1}{2} \leq \frac{P_i}{O_i} \leq 2 \ \vee \ (|P_i| \leq \delta_a \wedge |O_i| \leq \delta_a), \\ 0 & \text{otherwise,} \end{cases} \tag{4}$$

$$q = \frac{1}{N}\sum_i n_i \ , \ n_i = \begin{cases} 1 & \text{if } \left| \frac{P_i - O_i}{O_i} \right| \leq \delta_r \ \vee \ |P_i - O_i| \leq \delta_a, \\ 0 & \text{otherwise,} \end{cases} \tag{5}$$

$$R^2 = \left( \frac{1}{N}\frac{1}{\sigma_P \sigma_O}\sum_i (P_i - \overline{P})(O_i - \overline{O}) \right)^2, \tag{6}$$

$$FB = 2\frac{\overline{O} - \overline{P}}{\overline{O} + \overline{P}}, \tag{7}$$

$$MG = \exp\left( \overline{\ln(\widetilde{O_i})} - \overline{\ln(\widetilde{P_i})} \right) \ \text{with} \ \widetilde{\varphi} = \max(\varphi, \delta_a), \tag{8}$$

$$NMSE = \frac{1}{N}\sum_i \frac{(P_i - O_i)^2}{\overline{P}\,\overline{O}}, \tag{9}$$

$$VG = \exp\left( \overline{\left( \ln(\widetilde{O_i}) - \ln(\widetilde{P_i}) \right)^2} \right) \ \text{with} \ \widetilde{\varphi} = \max(\varphi, \delta_a), \tag{10}$$

with $O_i$ being the observed (wind-tunnel), $P_i$ the predicted (PALM) measurements, $\delta_r$ the relative deviation threshold, $\delta_a$ the absolute deviation threshold and $N$ the total number of measurements; the overline denotes an average over all measurements and $\sigma_P$ and $\sigma_O$ are the standard deviation of $P$ and $O$, respectively. The deviation thresholds were set to $\delta_r = 0.25$ for all variables as recommended by VDI (2005) and $\delta_a$ as given in Tab. 1.

The validation metrics were calculated for the wind velocities $u$, $v$ and $U_{\text{hor}}$, their variance $\sigma_u^2$ and $\sigma_v^2$, and the turbulence intensities $I_u$ and $I_v$ that are defined as the standard deviation divided by the mean horizontal wind speed. Results are listed in Tab. 1. In general, all validation metrics are close to their ideal values indicating a very high agreement between both experiments. The largest deviation between both experiments is apparent for $v$ where both $FAC2$ and $q$ give the lowest values. However, $q$ is still within the acceptable range of $q \geq 0.66$ defined by VDI (2005). The metrics also reflect the above-mentioned findings of PALM underestimating the mean wind speed. Both, $FB$ and $MG$ indicate an about $5\%$ underestimation ($MG = 1.5$). Also the underestimation of $\sigma_v$ visible in Fig. 10 is represented by $MG = 1.2$ indicating about $20\%$ lower $\sigma_v$ for the

**Table 1.** Calculated validation metrics for different variables. The right-most column gives the respective value of a perfect match between simulation and observation.

| metric | $u/u_{\mathrm{ref}}$ | $v/u_{\mathrm{ref}}$ | $U_{\mathrm{hor}}/u_{\mathrm{ref}}$ | $\sigma_u^2/u_{\mathrm{ref}}^2$ | $\sigma_v^2/u_{\mathrm{ref}}^2$ | $I_u$ | $I_v$ | ideal |
|--------|------|------|------|------|------|------|------|------|
| $FAC2$ | 0.98 | 0.73 | 1 | 0.98 | 0.98 | 1 | 1 | 1 |
| $q$ | 0.91 | 0.70 | 0.96 | 0.82 | 0.79 | 0.93 | 0.91 | 1 |
| $R^2$ | 0.97 | 0.87 | 0.96 | 0.57 | 0.55 | 0.83 | 0.85 | 1 |
| $FB$ | - | - | 0.03 | -0.06 | 0.19 | -0.08 | 0.03 | 0 |
| $MG$ | - | - | 1.05 | 0.95 | 1.2 | 0.93 | 1.04 | 1 |
| $NMSE$ | - | - | 0.01 | 0.07 | 0.21 | 0.05 | 0.07 | 0 |
| $VG$ | - | - | 1.01 | 1.05 | 1.08 | 1.02 | 1.02 | 1 |
| $\delta_a$ | 0.025 | 0.025 | 0.025 | 0 | 0 | 0 | 0 | |

PALM simulation. However, all metrics lie well within the margins reported by Hanna et al. (2004) for an acceptable performing model which are $FAC2 > 0.5$, $|FB| < 0.3$, $0.7 < MG < 1.3$, $NMSE < 4$, and $VG < 1.6$.

Hertwig et al. (2017) recommend to also evaluate the shape parameters of the wind speed distributions of $u$ and $v$ when comparing LES and wind-tunnel results. Therefore, the skewness $\gamma$ and the excess kurtosis $\beta$ are compared and shown in Fig. 11. Between both experiments, $\gamma_u$ mostly agrees and shows either symmetrical distributions ($\gamma_u \approx 0$) or a positive skew. For $v$, distributions tend to have a lower skewness in PALM than in the wind-tunnel measurements. Also, $\beta_v$ is smaller meaning less peaked distributions. This is also related to the higher roughness as this produces a wider spread of the distribution with a less pronounced peak resulting in lower $\beta$ and (in case of a positive average as is the case here) $\gamma$. Again, this is more pronounced in the span-wise wind component $v$.

The higher roughness and enhanced turbulence leads to a less correlated flow where length scales are also reduced. Figure 12 displays the comparison of length scales $L_u$ and $L_v$ of the $u$ and $v$ component, respectively. Length scales are calculated based on the integral time scale $T$:

$$L_\varphi = T_\varphi u_{\mathrm{ref}}, \tag{11}$$

where $T$ is calculated using the auto-correlation function $R_a$:

$$\int_0^{T_\varphi} R_{a,\varphi}(t_l) dt_l = e^{-1} \text{ with } R_{a,\varphi}(t_l) = \frac{1}{\sigma_\varphi^2} \overline{(\varphi(t) - \overline{\varphi}(t))(\varphi(t+t_l) - \overline{\varphi}(t+t_l))} \text{ for } \varphi \in \{u,v\}, \tag{12}$$

with $t_l$ being the time lag.

Most striking are the considerably lower values of $L_v$ within the PALM simulation. However, most data points still lie within the factor-of-two margins with $FAC2(L_v) = 0.8$. For $L_u$, also lower values tend to be underestimated while higher values tend to be overestimated.

In the following, we compare vertical profiles of various quantities As many measurement stations showed nearly identical behaviour within their vertical profiles, we limit the discussion to three stations: 4, 11, and 7. These are chosen to represent a good, an average and a relatively poor agreement, respectively, between PALM and wind-tunnel measurements.

Figure 13 shows vertical profiles of $U_{hor}$, $d$, $u$, $v$, as well as turbulence intensity $I$, skewness $\gamma$, excess kurtosis $\beta$ and length scale $L$ for $u$ and $v$ measured at station 7. Error bars show the standard deviation of $u$ and $v$ measurements. The blue shaded area shows the range of values of the neighbouring grid points within PALM at that measurement station.

Station 7 is situated at the opening of a street canyon within the lee of a building edge (see Fig. 3). Because the surrounding building walls are not aligned with the PALM grid, the building edge has a different shape within the simulation than it had

in the wind-tunnel experiment. This then causes the formation of a slightly enlarged corner vortex. As a result, wind speed is reduced and turbulence increased compared to the wind-tunnel results as shown in Fig. 13. Also, $d$ is affected and deviates from the wind-tunnel results. Above the rooftop, which is situated between $26\,\mathrm{m}$ and $36\,\mathrm{m}$ in this area, $d$, $I_u$ and $I_v$ agree significantly better to the wind-tunnel measurements, because the effect of rougher building walls is mostly limited to the canopy layer. Due to higher turbulence, also $\beta_v$ is decreased indicating a less peaked distribution while lower $\gamma_u$ indicates

larger tails towards low $u$ values. Higher turbulence due to larger roughness causes $L$ to be shorter within the canopy as well. The higher roughness introduced by the rougher building walls causes higher vertical momentum flux and hence larger mean wind speed and length scales above the canopy layer. Similar behaviour can be found at station 20 (profiles not shown). There, the station is situated closely to a windward building corner. In that case, the blocking effect of the building is increased due to the broader building edge causing significant differences in wind direction distribution and mean wind speed (see Fig. 8 and

9).

Profiles not affected by corner flows or the blocking effect tend to agree better between PALM and wind-tunnel measurements. Station 11 is positioned at the centre of a street canyon (see Fig. 3) and profiles tend to agree significantly better within the canopy layer as shown in Fig. 14. Higher deviations between the experiments appear near the building roof level which again is between $26\,\mathrm{m}$ and $36\,\mathrm{m}$. The roofs of the surrounding buildings are not flat but have small structures which might not

be sufficiently resolved and lead to differences between the experiments. Results at stations 5, 6, 10, and 17 are comparable to those of station 11. Important to note is the large range of values at the surrounding grid points shown by the blue-shaded area within Fig. 14. Profiles vary significantly within a street canyon depending on the distance to the building walls. Hence, placing the measurements correctly within the simulation is very important if situated in close vicinity to buildings.

The less complex the building structures are, the better PALM reproduces the flow of the wind-tunnel experiment. Station 4

is situated at the leeward site of a fairly simply structured building (see Fig. 3). Profiles, displayed in Fig. 15, show nearly no differences between the two experiments. Only the influence of the rougher wall due to the building representation is noticeable producing slightly larger turbulence and a less peaked distribution of $u$ and $v$ within the canopy layer.

Finally, turbulence spectra in form of the spectral energy density $S$ are compared between PALM and wind-tunnel measurements. Figure 16 shows the dimensionless energy spectra for station 4 (left panel) and 11 (right panel) at different heights $z$.

Spectra measured at station 7 are comparable to those of station 11 and are therefore not shown. Note that the covered range of frequencies $f$ differ between PALM and wind-tunnel measurements as the sampling rate of the measurements and the mea-

sured time interval vary between the PALM and wind-tunnel experiment. However, results of both experiments still overlap over a large range of frequencies.

In general, spectra for $u$ and $v$ coincide to a high degree between PALM and wind-tunnel measurements at all heights. The inertial range of the turbulence spectrum is clearly visible within both experiments at $75\,\text{m}$ height (above the canopy layer) at both stations (Figs. 16a and b). The normalized energy spectrum decays with roughly $fS \propto f^{-\frac{2}{3}}$ following Kolmogorov's theory. At high frequencies, spectra of the PALM measurements strongly decay, which is related to numerical dissipation and is a typical behaviour of LES models using high-order differencing schemes (e.g., Glendening and Haack, 2001; Kitamura and Nishizawa, 2019).

At rooftop height (Figs. 16c and d), PALM's spectra are shifted towards higher frequencies compared to those of the wind tunnel at the same height. Hence, PALM simulates smaller-scale turbulence at these heights, which might be related to higher roughness and further fosters the above-mentioned findings from the profile analysis. The brick-like representation of the buildings introduces additional roughness causing smaller turbulence elements and hence a shift of the energy spectrum to higher frequencies. The effect is more pronounced for station 4 than for station 11 which, however, might also be related to the more distinct maximum and hence better visibility of a shift at station 4.

Spectra close to the surface agree better between PALM and the wind tunnel measurements. Though, due to the limited measurement frequency and very small turbulent structures at the surface, the inertial range is not covered by the wind-tunnel measurements at $3\,\text{m}$ height. This height corresponds to the third grid level above the surface of the PALM simulation and therefore it can be truly expected that the inertial range is only poorly resolved. Comparing the measured spectra to the theoretical decay of $f^{-\frac{2}{3}}$, the inertial range is indeed hardly represented within the data.

Results within the vicinity of the measurement stations could be improved by utilizing PALM's self-nesting feature (Hellsten et al., 2020). This allows to use a higher grid resolution within specific areas of the model domain. We recommend that future simulations should try using this feature for areas requiring high resolution.

## 4    Conclusions

In this study, we analyzed PALM's capability to simulate a complex flow field within a realistic urban building array. Simulation results were compared with measurements done at the EWTL facility at the University of Hamburg, Germany. The aim was to evaluate the dynamic core of the newest version 6.0 of PALM, which underwent significant code-changes in recent model development.

The comparison of PALM results with the wind tunnel data proves that PALM is capable to correctly simulate a neutrally stratified urban boundary layer produced by a realistic complex building array. Measurements from both experiments were compared at several different positions throughout the building array at non-obstructed locations, at the windward and leeward site of buildings as well as within street canyons and at intersections. Overall, the PALM results displayed very good agreement with the corresponding wind-tunnel measurements in regards to wind speed and direction as well as turbulence intensity. Validation metrics as proposed by Schatzmann et al. (2010) were all within the acceptable range.

However, it was found that PALM underestimates wind speed and overestimate turbulence close to the ground and building surfaces. Differences were most pronounced in the span-wise wind velocity component. Such discrepancies were also reported recently by Paas et al. (2020) when comparing PALM simulations to real-world measurements. These differences can be ascribed partly to an overestimation of roughness mainly introduced by the step-wise representation of the buildings onto PALM's rectilinear grid. This causes building walls not aligned with the grid to appear significantly rougher resulting in lower wind speed and higher turbulence close to walls and especially in the vicinity of building corners. Also, the used roughness length of $z_0 = 0.01\,\mathrm{m}$ might be larger than the actual surface roughness as present in the wind-tunnel experiment causing the highest difference of mean wind speed ($-9\,\%$) at the lowest analysis height.

To a lesser degree, also the miss-match in measurement height is found to be responsible for a difference in mean wind speed. Due to PALM's staggered Arakawa C grid, output was not available at the exact same position as in the wind-tunnel experiment but was shifted half a grid spacing ($0.5\,\mathrm{m}$) downwards accounting for up to $3\,\%$ difference in wind speed at the lowest grid level.

If $z_0$ is unknown, this can certainly produce differences between PALM and reference data close to surfaces. More importantly, however, is a good representation of building structures. If the focus lies on flow features in close vicinity to buildings, the most important buildings should be aligned with the simulation grid. Also, a high grid resolution is recommended to represent structures as close to the reference as possible. To achieve this, PALM's nesting feature could be utilized in order to cope with increasing computational demand if the grid size is reduced (Hellsten et al., 2020). A higher resolution has the additional benefit of reducing errors introduced by shifting locations of measurements if PALM results are compared against reference data which otherwise can cause deviations close to building structures where large gradients can lead to significant differences in results (as can be seen, e.g., by the range of profiles at station 11, Fig. 14). In a future release of PALM, an immersed boundary condition is going to be available (Mittal and Iaccarino, 2005, e.g.). This will then mitigate the increased roughness effect introduced by the step-wise representation of building walls not aligned with the rectilinear grid.

Lastly, we would like to give some general advice for the setup preparation. In the present study, we experienced that input data must always be checked with very high caution. Especially large building data sets might contain errors and false building heights or missing/displaced buildings, which are more difficult to spot than in setups with a limited number of buildings. This is, of course, of utmost importance for the area of interest, but also the upwind region requires proper verification as it directly affects the analysis area. Also, when comparing to other experiments like real-world or wind-tunnel measurements, positioning of the measurements must be thoroughly checked as mentioned by Paas et al. (2020). This is also true for positioning virtual measurements within the PALM domain. At positions with highly complex wind fields, it can make a large difference for the results if measurement positions are off by only a single grid point. This of course depends strongly on the grid spacing used and will be most relevant when using relatively coarse grids.

This study focused only on a single but the most essential part of PALM, the dynamic core. However, a full validation of the entire model requires additional studies focusing on the other model parts like the radiation module, the chemistry module or the land-surface module to mention only a few. Some of these are already validated (Resler et al., 2017; Kurppa et al., 2019; Fröhlich and Matzarakis, 2019; Gehrke et al., 2020), others will follow in future publications.

*Code and data availability.* The PALM model system is freely available from http://palm-model.org and distributed under the GNU General Public License v3 (http://www.gnu.org/copyleft/gpl.html). The model source code of version 6.0 in revision 3921, used in this article, is archived on the Research Data Repository of the Leibniz University of Hannover (Gronemeier et al., 2020b) as well as input data and measurement results presented in this paper in combination with plotting scripts to reproduce the presented figures (Gronemeier et al., 2020a).

*Author contributions.* BL and KS created the wind-tunnel setup. KS conducted the wind-tunnel measurements and analysis with supervision of FH and BL. TG, SR and BM created the simulation setup. TG carried out the simulations and precursor test simulations with supervision of SR. All authors took part in data analysis of the comparison. TG compiled the manuscript with contributions by all coauthors.

*Competing interests.* The authors declare that no competing interests are present.

*Acknowledgements.* We like to thank everyone who helped to conduct the experiments and helped by writing software for the analysis of 380 the results (including data preparation, model building, and code bug-fixing). We also like to thank the anonymous referees who helped to significantly improve the script with their valuable comments. This study was part of the [UC]$^2$ project that is funded by the German Federal Ministry of Education and Research (BMBF) within the framework 'Research for Sustainable Development' (FONA; www.fona.de). It was conducted in collaboration of the two sub-projects MOSAIK (funding code: 01LP1601) and 3DO (funding code: 01LP1602). The authors would like to thank Wieke Heldens and Julian Zeidler at the German Aerospace Center (DLR) for the support during the project and 385 especially for preparing the building data used within this study. All simulations were carried out on the computer clusters of the North-German Supercomputing Alliance (HLRN; www.hlrn.de). Data analysis was done using Python.

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

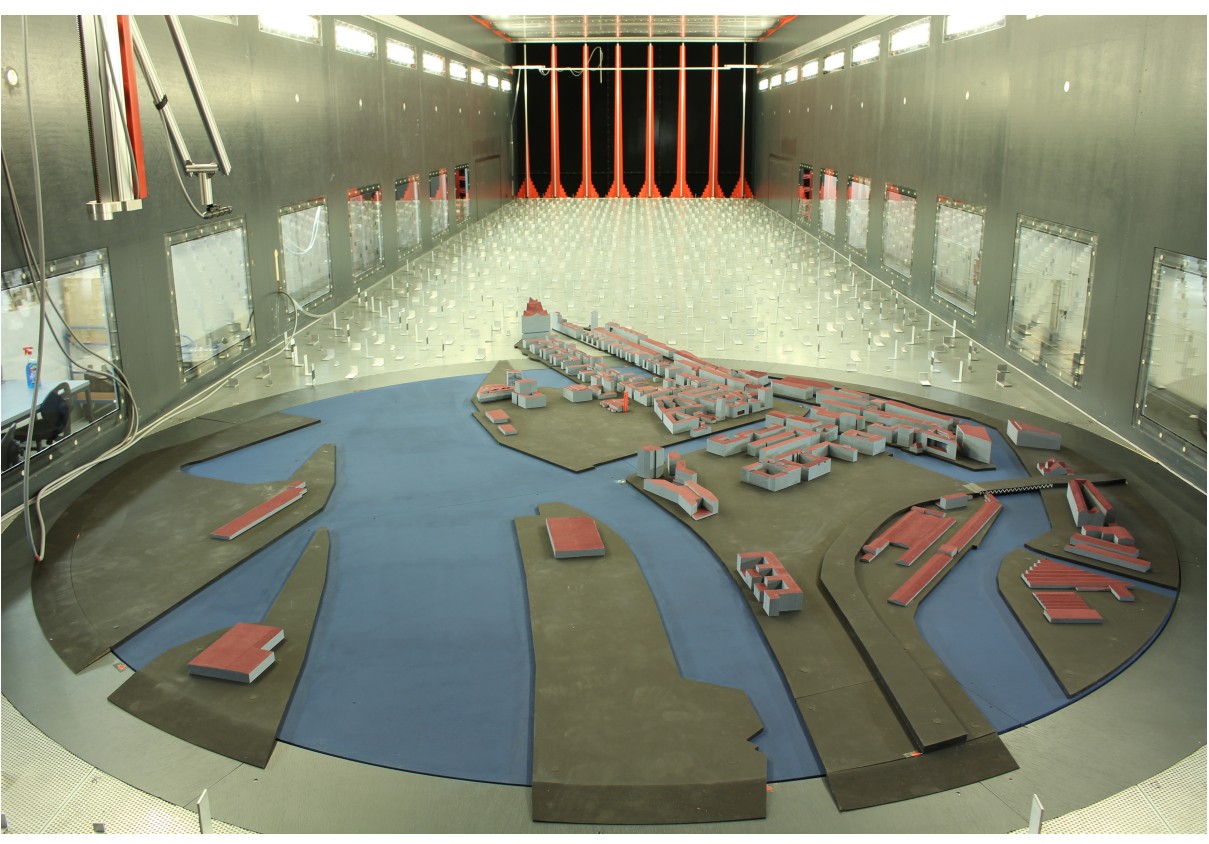

**Figure 1.** Photograph of the building setup within the wind-tunnel facility 'WOTAN' for an approaching flow of $290°$. Please note that contrary to the depicted orientation, an approaching flow from $110°$ was used within this study.

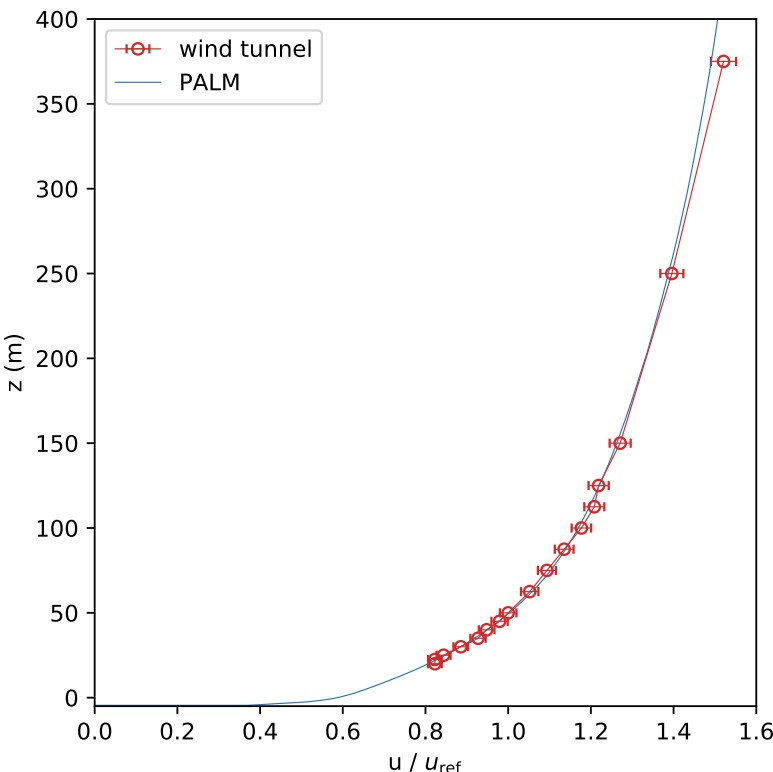

**Figure 2.** Mean profiles of the approaching flow for the wind-tunnel experiment and the PALM simulation normalized with the reference velocity $u_{\text{ref}} = u(z = 50\,\text{m})$. Note that $z = 0\,\text{m}$ is defined at street-level height while the lowest level within both experiments was at $z = -5\,\text{m}$, which is the water-level height.

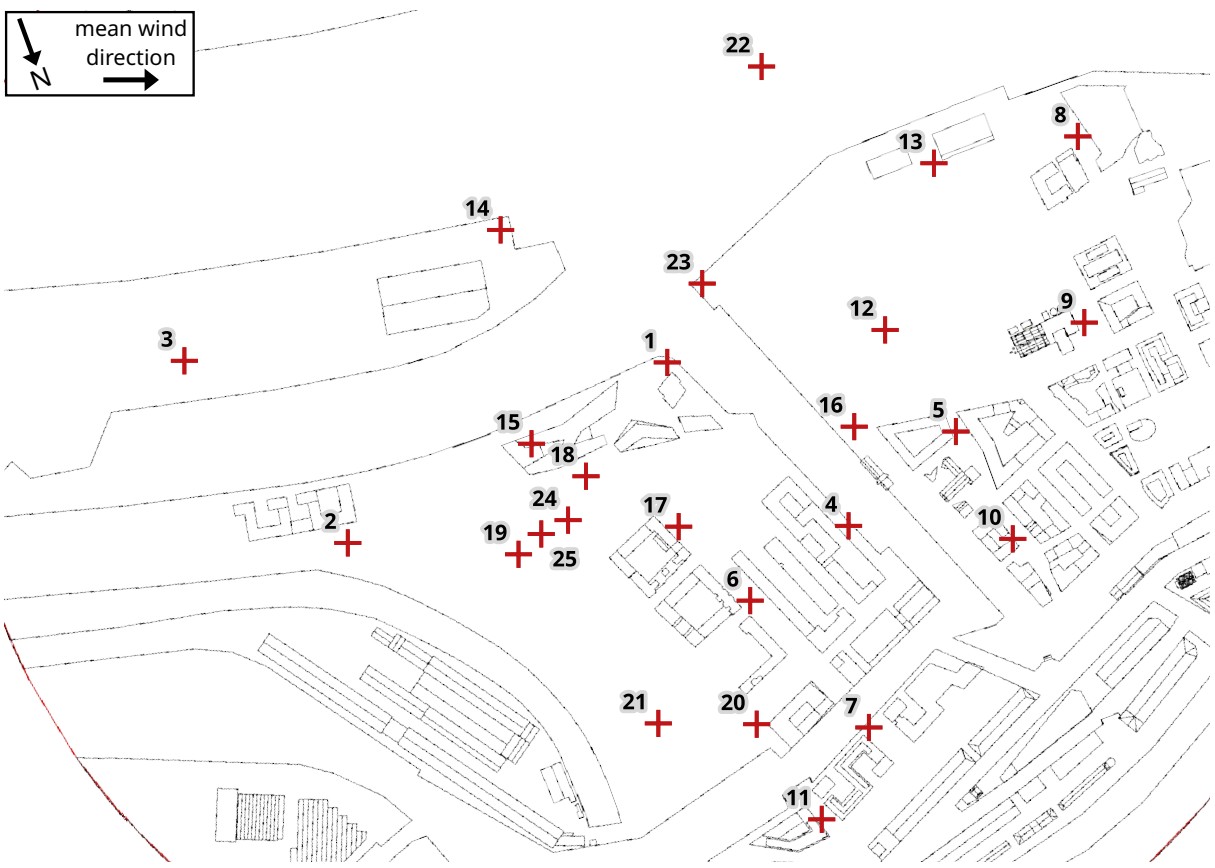

**Figure 3.** Building layout used in the wind-tunnel experiment. Measurement locations are marked and labeled by their respective number.

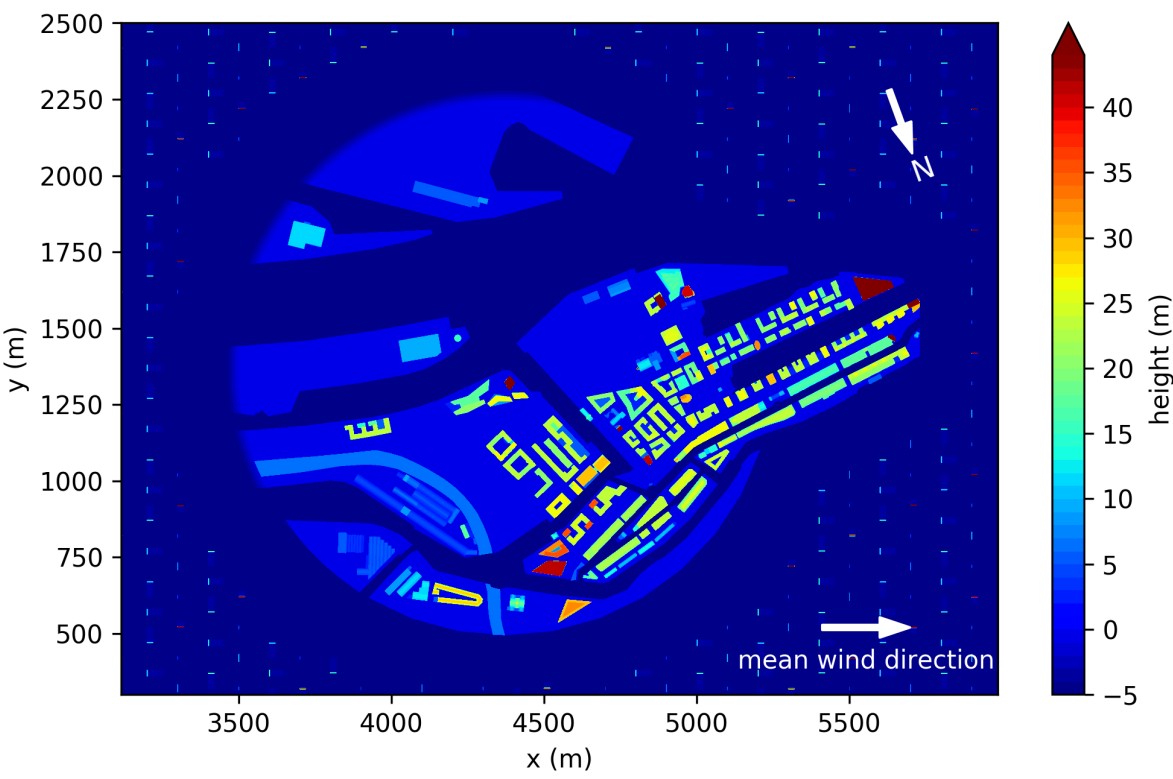

**Figure 4.** Building layout and heights as used in the PALM simulation. The $x$-direction is oriented to follow the mean wind direction. The total domain size is $6000\,\mathrm{m}$, $2880\,\mathrm{m}$, and $601\,\mathrm{m}$ in $x$-, $y$-, and $z$-direction, respectively. Note that $z = 0\,\mathrm{m}$ is defined at street-level height while the lowest level within both experiments was at $z = -5\,\mathrm{m}$, which is the water-level height.

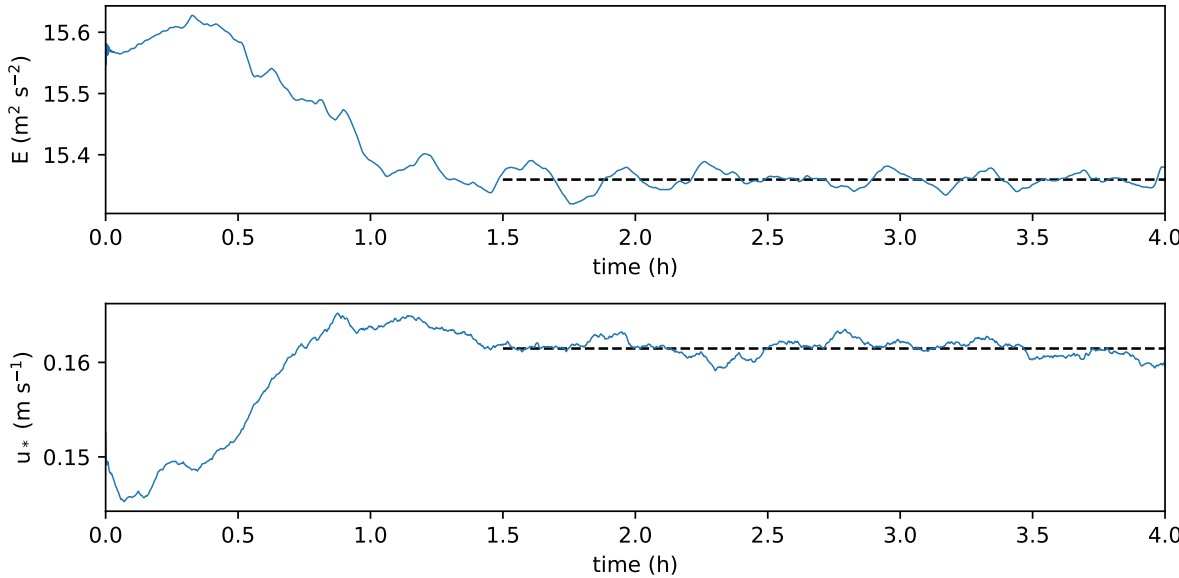

**Figure 5.** Time series of the total kinetic energy $E$ and the friction velocity $u_*$ of the PALM simulation.

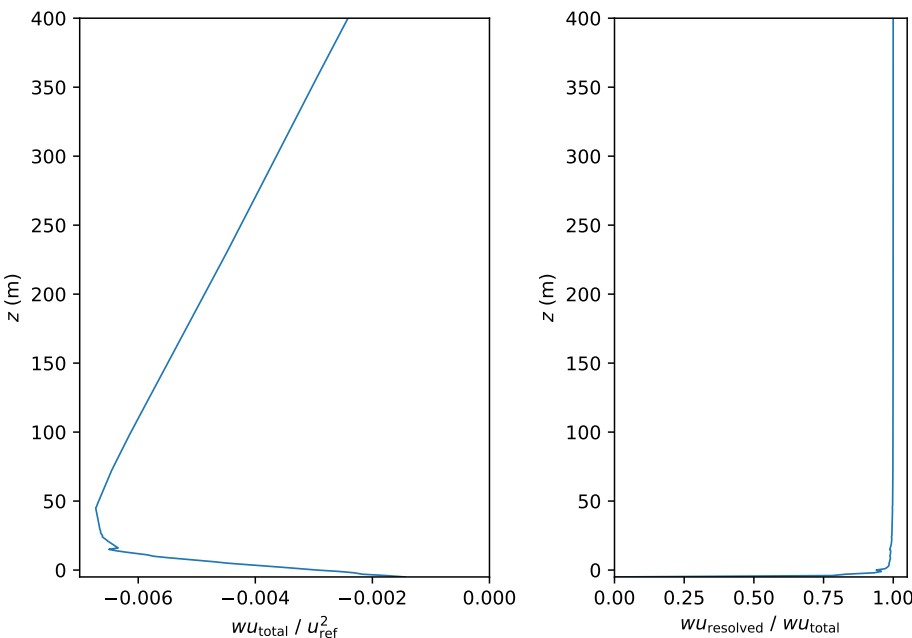

**Figure 6.** Mean profile of the vertical momentum flux and the ratio between resolved and total flux, averaged over the entire domain of the PALM simulation. Please note the two different horizontal scales for momentum flux (bottom scale) and flux ratio (top scale).

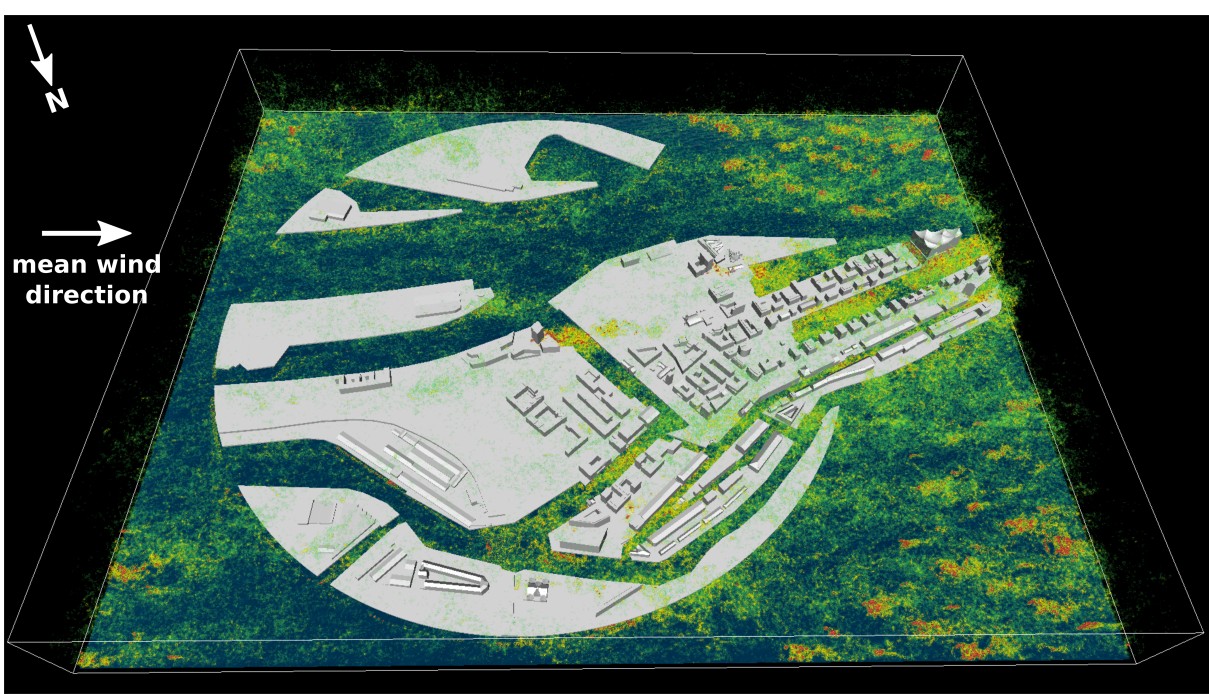

**Figure 7.** View of the volume-rendered instantaneous turbulence structures above the building array. Turbulence is visualized using the magnitude of the three-dimensional vorticity. Green and red colour show low and high values, respectively. Image was rendered using VAPOR (Li et al., 2019, www.vapor.ucar.edu), the background image was designed by freepic.diller / Freepik (http://www.freepik.com).

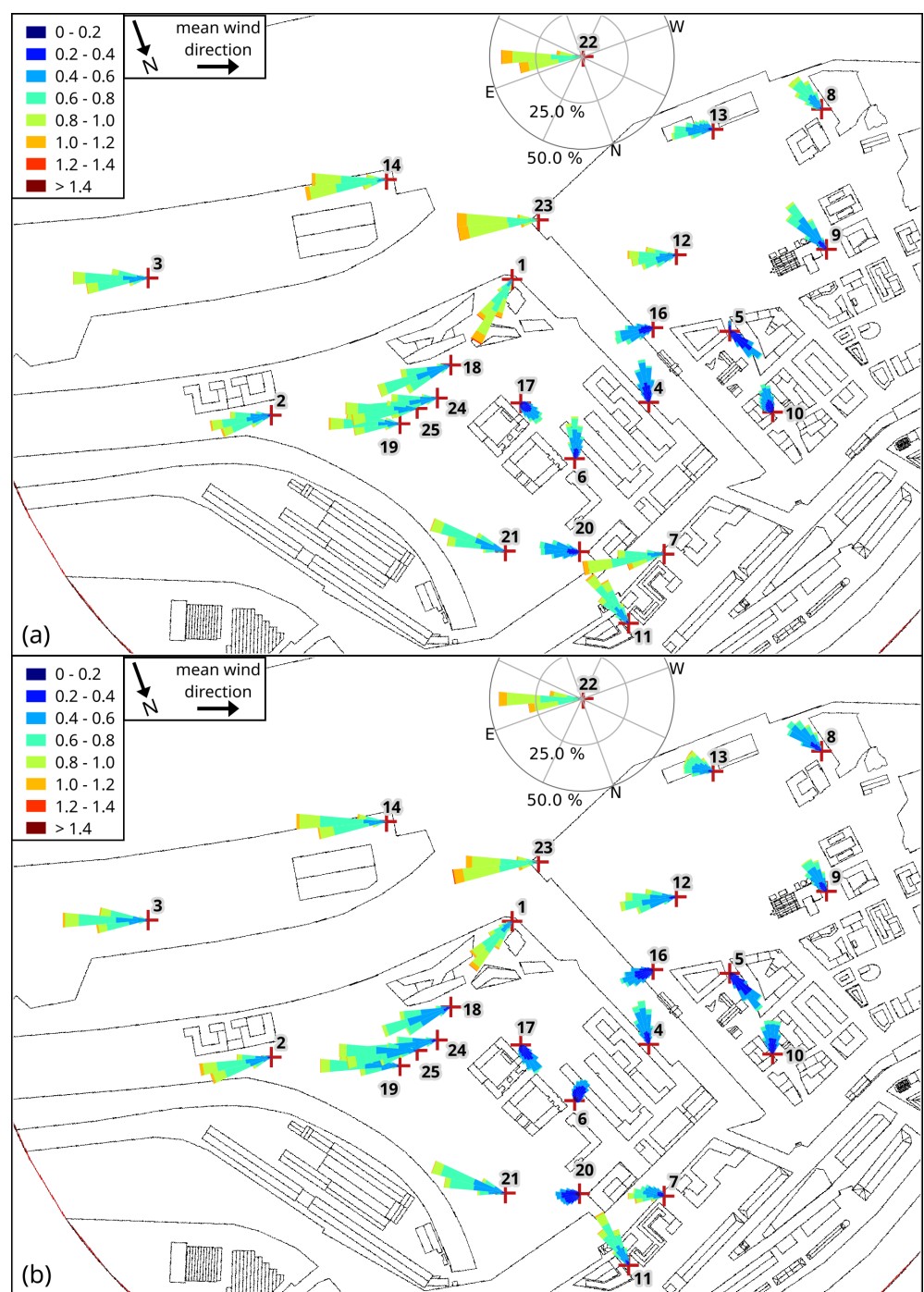

**Figure 8.** Wind-speed distribution (wind rose plots) at all measurement stations for (a) wind-tunnel measurements and (b) the PALM simulation at about $3\,\mathrm{m}$ height above street level (wind tunnel: $3\,\mathrm{m}$, PALM: $2.5\,\mathrm{m}$). Axes are only shown for a single station for better overview, but all wind distributions are scaled equal. Numbers indicate the station number.

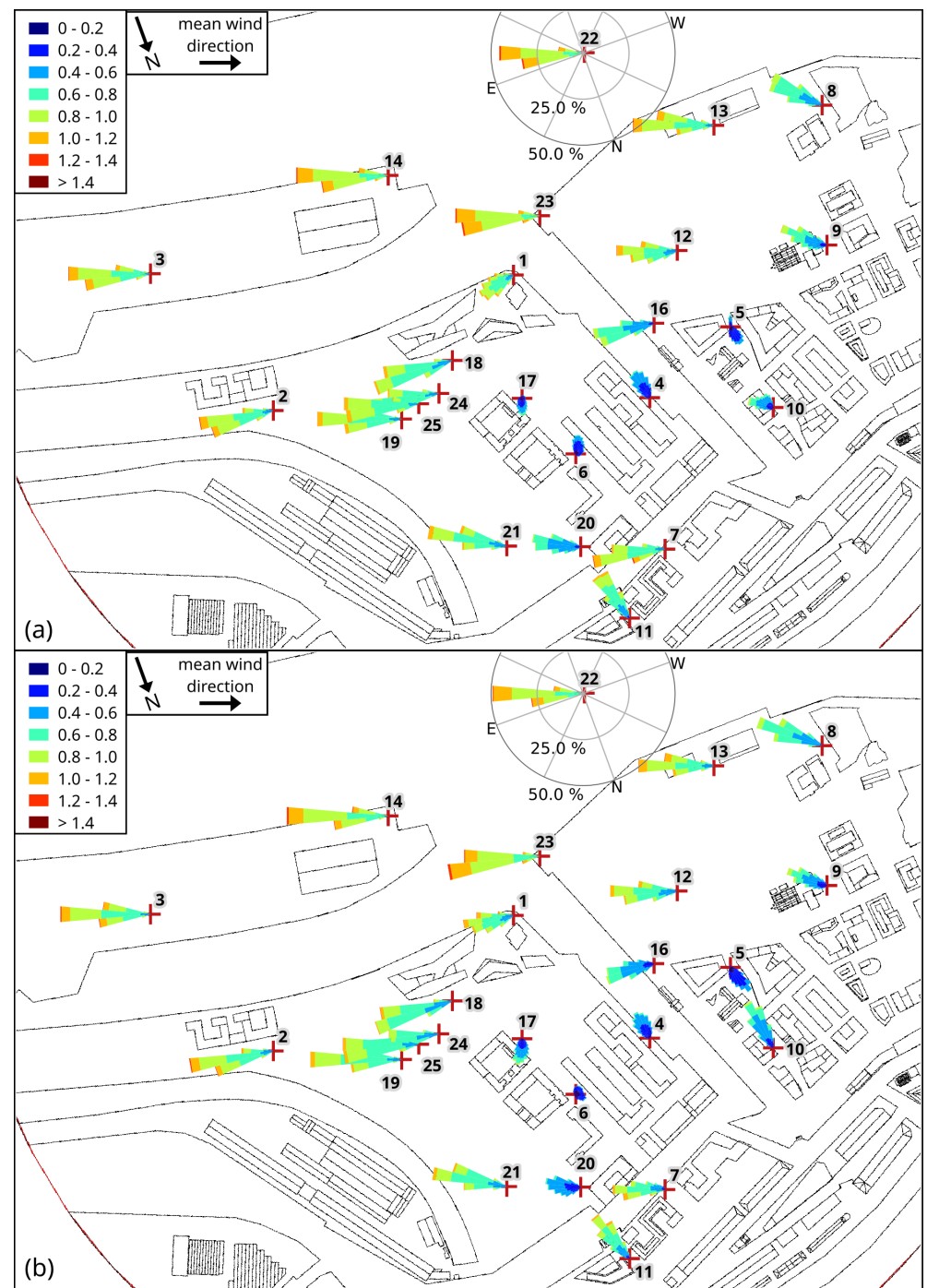

**Figure 9.** Wind-speed distribution (wind rose plots) at all measurement stations for (a) wind-tunnel measurements and (b) the PALM simulation at about $10\,\mathrm{m}$ height above street level (wind tunnel: $10\,\mathrm{m}$, PALM: $9.5\,\mathrm{m}$). Axes are only shown for a single station for better overview, but all wind distributions are scaled equal. Numbers indicate the station number.

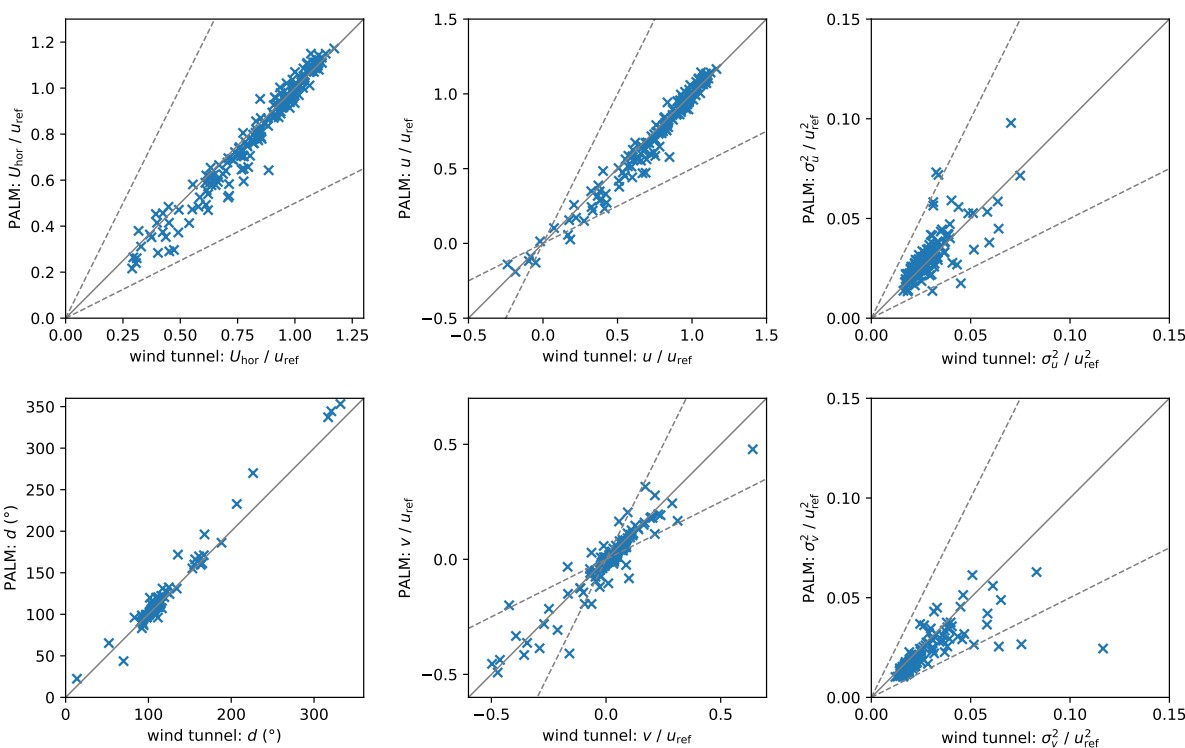

**Figure 10.** Scatter plots of wind tunnel and PALM measurements of horizontal wind speed $U_{\mathrm{hor}}$, wind direction $d$, wind-velocity components $u$ and $v$, and their variance $\sigma_u^2$ and $\sigma_v^2$ for all 25 measurement stations and all heights (173 data pairs in total). Solid lines indicate perfect agreement, dashed lines indicate the area between a deviation factor of 0.5 and 2.

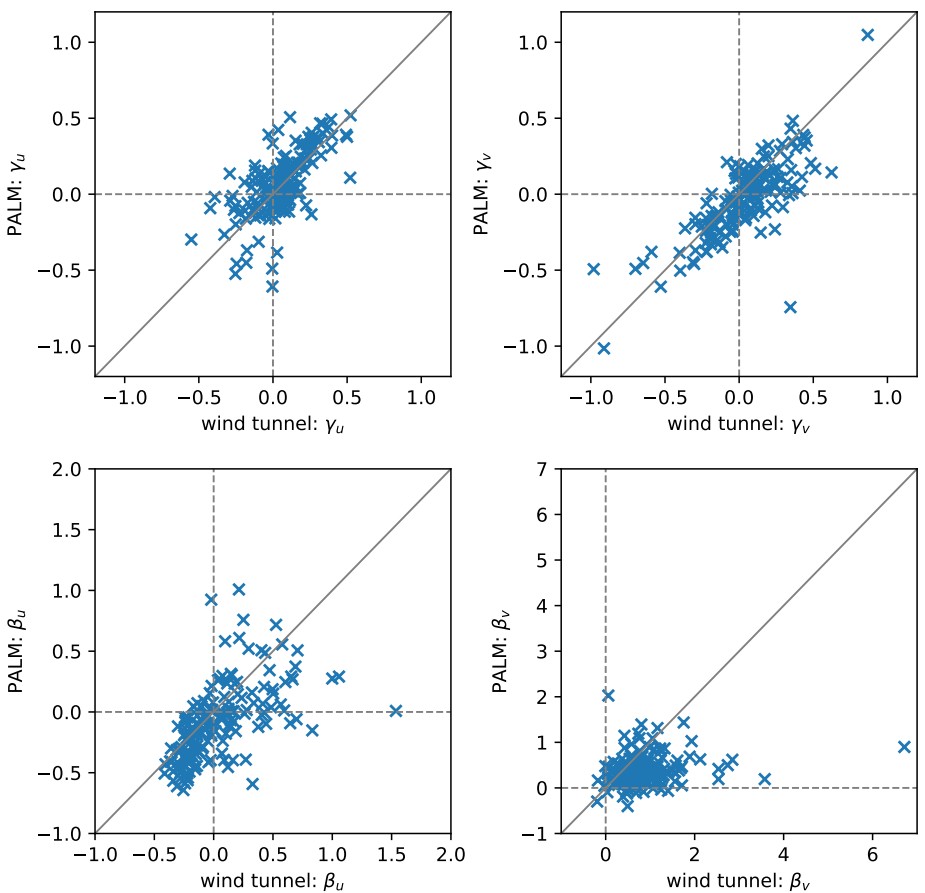

**Figure 11.** Scatter plots of wind tunnel and PALM measurements of skewness $\gamma$ and excess kurtosis $\beta$ of the horizontal wind velocity components for all 25 measurement stations and all heights (173 data pairs in total). Solid lines indicate perfect agreement.

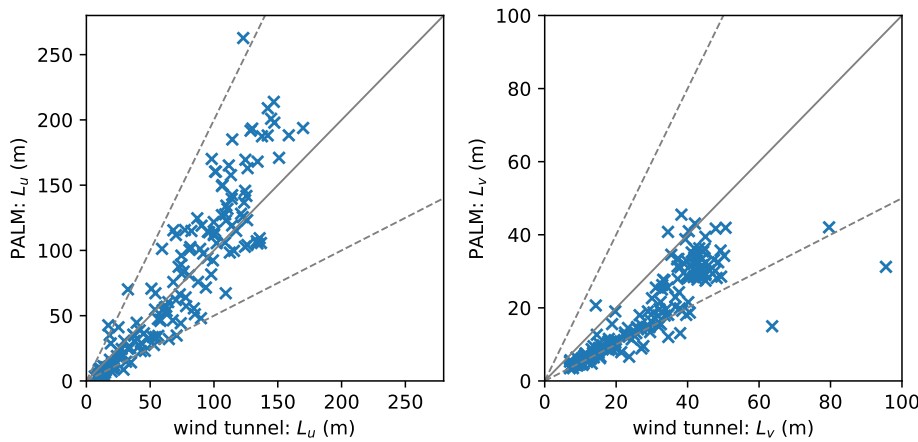

**Figure 12.** Scatter plots of wind tunnel and PALM measurements of the length scales $L_u$ and $L_v$ of the velocity components $u$ and $v$, respectively, for all 25 measurement stations and all heights (173 data pairs in total). Solid lines indicate perfect agreement, dashed lines indicate the area between a deviation factor of 0.5 and 2.

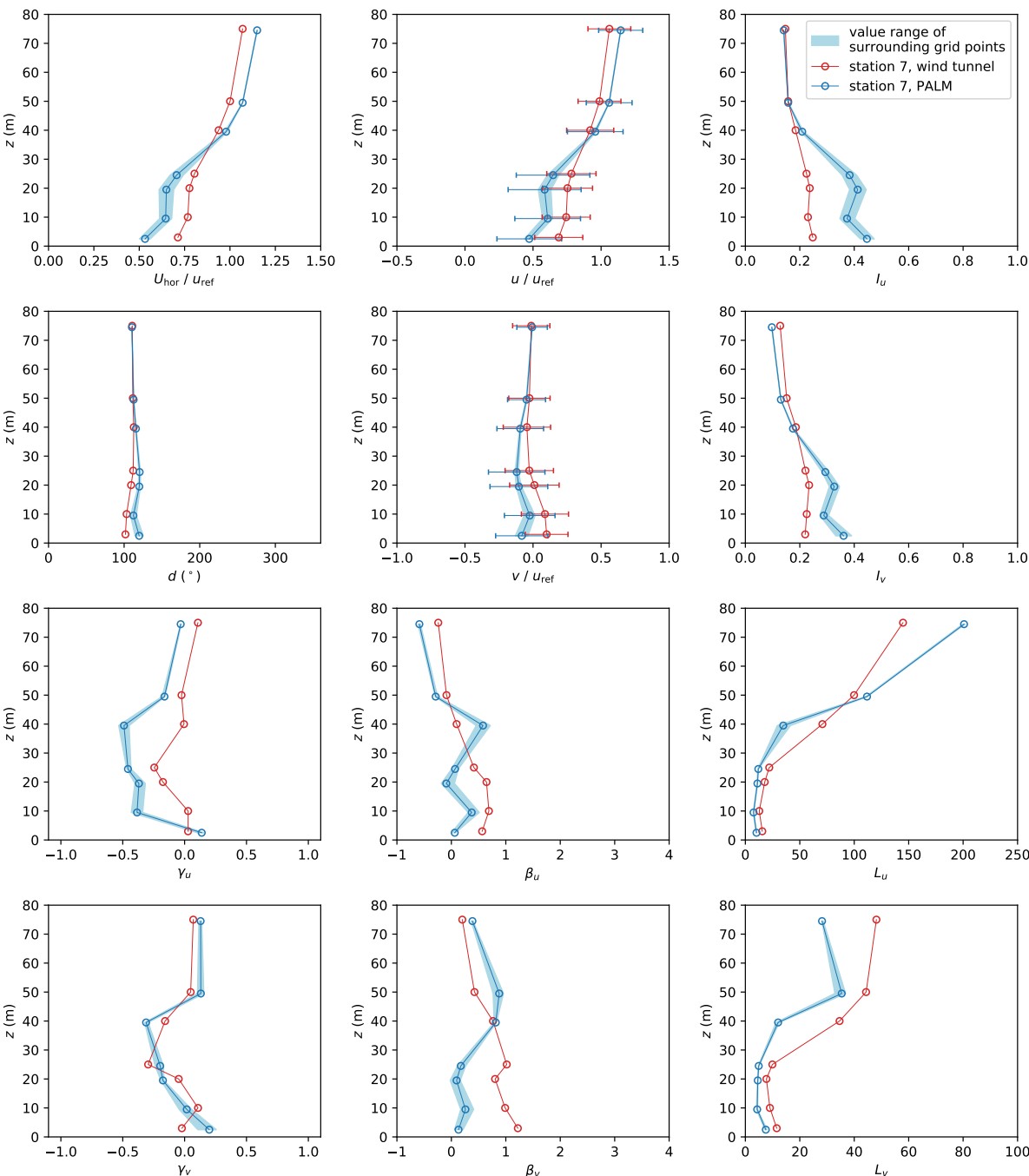

**Figure 13.** Profiles of mean horizontal wind speed $U_{\text{hor}}$, wind direction $d$, wind components $u$ and $v$, as well as turbulence intensity $I$, skewness $\gamma$, excess kurtosis $\beta$ and length scale $L$ of both wind velocity components $u$ and $v$ at measurement station 7. Error bars denote the standard deviation of the respective quantity. Note that $z = 0$ denotes street-level height.

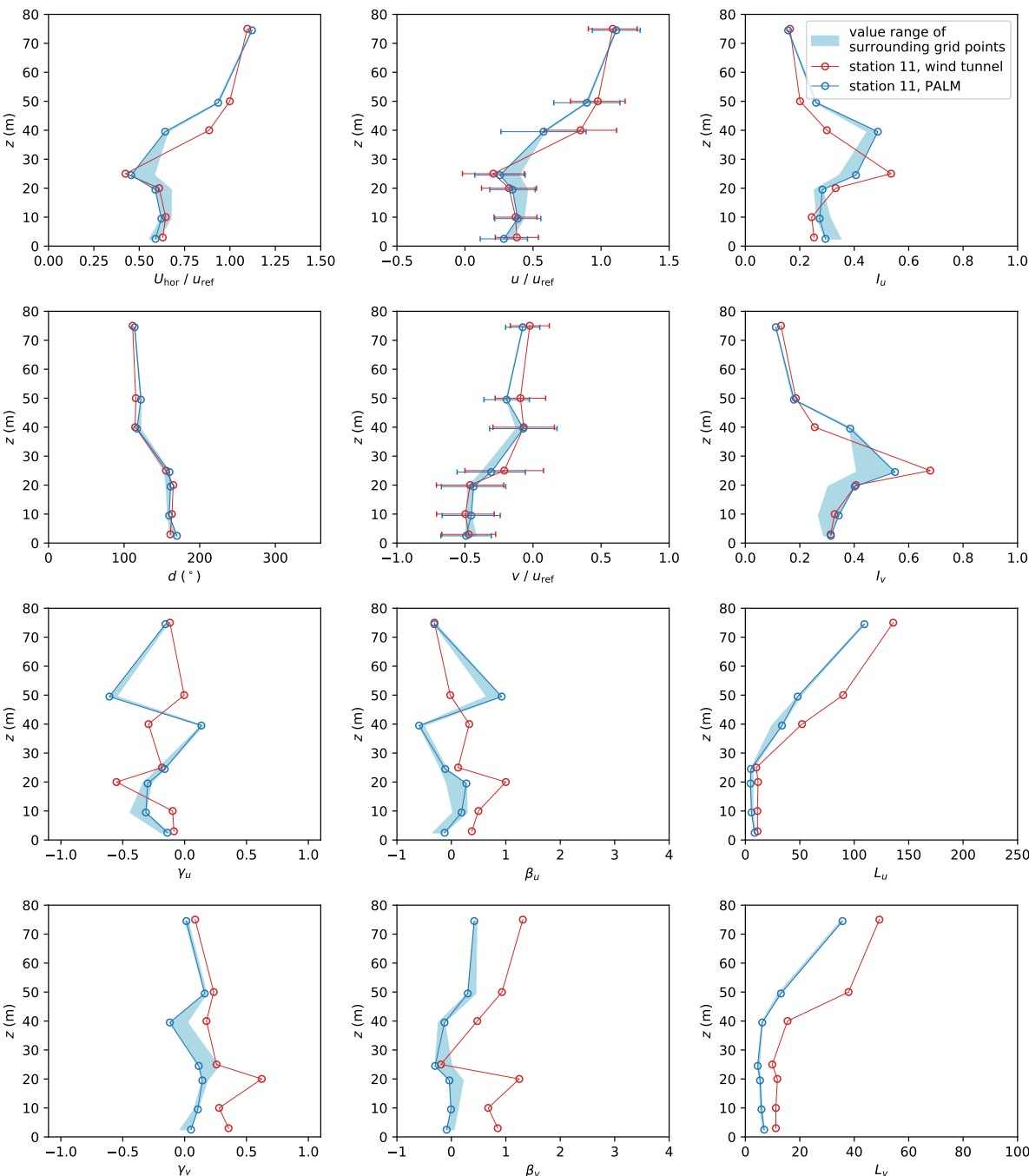

**Figure 14.** Profiles of mean horizontal wind speed $U_{hor}$, wind direction $d$, wind components $u$ and $v$, as well as turbulence intensity $I$, skewness $\gamma$, excess kurtosis $\beta$ and length scale $L$ of both wind velocity components $u$ and $v$ at measurement station 11. Error bars denote the standard deviation of the respective quantity. Note that $z = 0$ denotes street-level height.

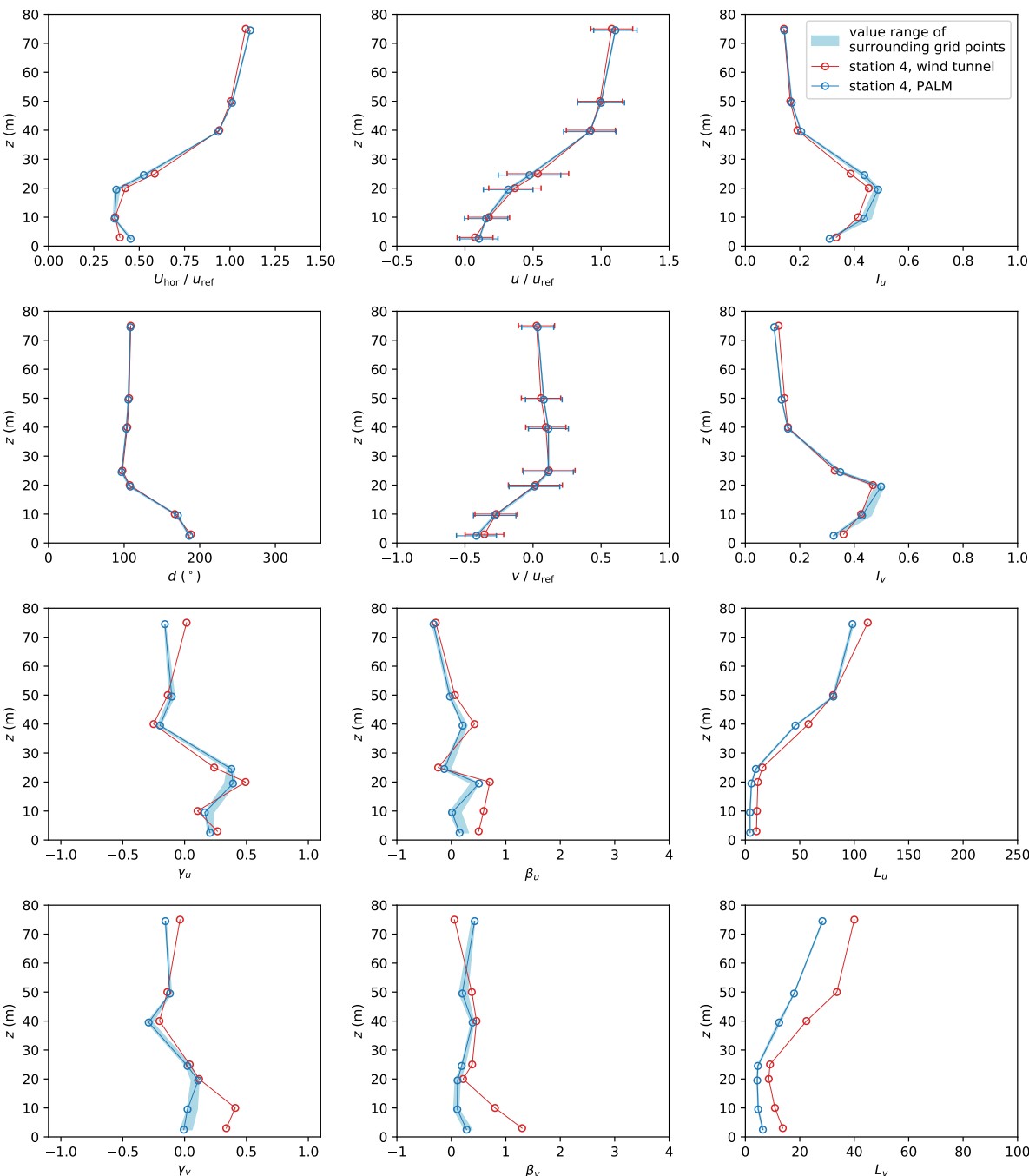

**Figure 15.** Profiles of mean horizontal wind speed $U_{\mathrm{hor}}$, wind direction $d$, wind components $u$ and $v$, as well as turbulence intensity $I$, skewness $\gamma$, excess kurtosis $\beta$ and length scale $L$ of both wind velocity components $u$ and $v$ at measurement station 4. Error bars denote the standard deviation of the respective quantity. Note that $z = 0$ denotes street-level height.

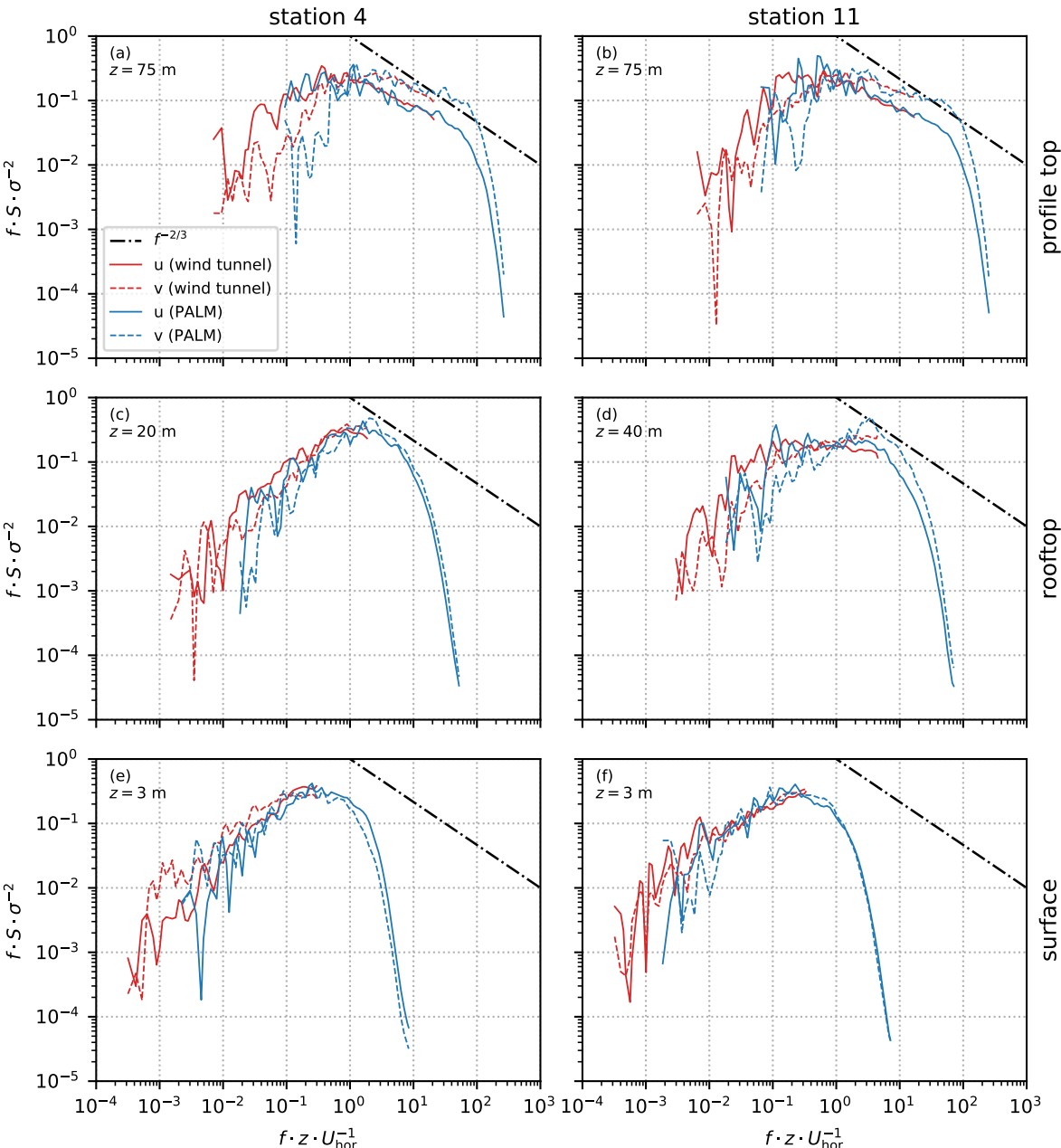

**Figure 16.** Spectral energy density $S$ for $u$ and $v$ at station 4 (left) and station 11 (right) at profile top (a, b), rooftop height (c, d), and near the surface (e, f). $S$ is normalized by multiplying with the frequency $f$ and dividing by the variance $\sigma^2$. For reference, the dash-dotted line shows a slope $f^{-\frac{2}{3}}$ indicating the slope of energy decay according to Kolmogorov's theory. Note that $z$ is given relative to street-level height.