# Peer review of "Evaluation of the Dynamic Core of the PALM Model System 6.0 in a Neutrally Stratified Urban Environment: Comparison between LES and Wind-tunnel Experiments"

_Geoscientific Model Development, 2020_

## Referee Comment (RC1) · Anonymous Referee #1 · 28 Sep 2020

General comments: This study has examined the representation of the turbulent flow characteristics reproduced by a newly updated CFD model, PALM6.0. The simulation results are compared with a wind tunnel experiment regarding the flow around a realistic building geometry in Hafencity. The results were generally well correspondence with the wind tunnel experiment, not only the simple mean wind speed but also the variance of the wind velocity and the spectral variation. This report will be useful for the users of this model. In addition, it also indicates some general issues in the simulation of the urban airflow. Therefore, this work is worth for publication. Besides, I still have some comments to be clarified before publication. One is about the model description. This study is motivated to analyze the model performance which is updated to version

6 of PALM. However, it is not sure if the core part of the model, which is related to the prognostic equation of the momentum (e.g. advection, diffusion, time-integration or wall boundary conditions, etc.) except the thermal effect, are updated from the previous version. Please describe more details about this point clearly. The results should also be more focused on the influence of the updated parts. Another point is the upwind condition. I think the results could be different near the upwind region if the inflow conditions are so different (Fig.2). However, the authors indicated that the observation points near the upwind region (e.g. station 2, 3, etc.) are almost same both in CFD and wind tunnel experiment. I am not sure why they are the same results irrespective of the different approach flows. Although it is fair to use the periodic boundary conditions in the streamwise direction, the effect of the inflow condition has to be more carefully examined (e.g. how far the direct influence of the inflow is observed?).

Specific comments: L7: "In the end, . . . " This could not be discussed from the materials shown in the present manuscript. L27: "drastic change . . ." I could understand there are many updates for the application parts but still not sure for the core prognostic parts. This is related to the general comment. L71: "25 locations . . ." How are these station chosen? For example, how the average of 25 vertical profiles becomes compared with the total horizontal average in the numerical simulation? L76: " we skip . . ." It has to be indicated here about which parts of the prognostic calculation were updated. L96: "1.54 m /s . . ." This is rather weak at the top of the boundary layer if this is in the real atmospheric scale. Are the present results really free from this main wind speed? L158, L167, . . .: "the approaching flow at a height of 50 m" "wind speed is 23 % less . . ." Please explain why the wind speed at this height is the reference. Since the correspondence of the magnitude of the vertical profile directly depends on this parameter, it needs justification to use this parameter as a representative velocity scale. L170: "located 0.5m lower" Why the values are not interpolated (e.g. linear interpolation, cubic spline, etc.) for the comparison? L240: "At roof top height" How is the effect of the mismatch of the local horizontal wind speed in the numerical simulation and the wind tunnel? I think the ratio of the building sizes, which will be related to the

peak wavelength at the roof level, and that of the local wind speed will be different for the numerical simulation and the wind tunnel experiment.

---

## Author Comment (AC1) · 15 Oct 2020

**Answers to the general comments**

We would like to thank the anonymous referee for his/her comments on our manuscript. It was commented that changes between the old and the new PALM version were just mentioned as "drastic" but not further described. The changes between PALM4.0 and PALM6.0 (version 5 does not exist) are manifold. A detailed description to the current state and the changes between both versions are given by Maronga et al. (2020). However, we agree that the reader should at least get an overview on the changes regarding the dynamic core of the model. The most significant changes within the code are more related to code structure and less to the physics. With the introduction of the land-surface and building surface models, the internal treatment of surfaces changed significantly. Several one-dimensional Fortran data structures were introduced to handle surface treatment. This also includes treatment of surfaces were neither the land-surface model nor the building-surface model is used as it is the case in our study. Also, further modularization of the code, i.e. reordering code parts like boundary treatment or turbulence closure into separate modules changed the overall code structure of the PALM model also affecting the model core. These changes should ideally have no effect on the simulation results but in reality might have small changes due to changes in computational order and rounding errors or even have changes due to fixed bugs which might have been not even noticed. Further changes which might have an effect on the results are the optimization of the multigrid solver which is also described by Maronga et al. (2020) and the calculation of the constant-flux layer which uses the Monin-Obukhov similarity theory (MOST) which was also used in previous versions of PALM but was also completely reworked for version 6.0. New code features, which were introduced in version 6.0 and which were used within this study, include the y-shift method and the virtual measurements module both of which we mentioned within our manuscript. We are going to modify our manuscript in order to briefly mention the changes between the old and the new PALM version.

The second general commend made was regarding the inflow condition. It is mentioned that although the approaching flow shows clear differences between PALM and the wind-tunnel data, profiles are said to match well for stations 2 and 3 which are positioned close to the windward edge of the analysis area. Differences between the approaching flow are due to the different setup of the upwind area between both experiments. While roughness elements are present within the wind-tunnel experiment, there was only a flat surface present within the PALM simulation as discussed in Sect. 2.2 in the manuscript. A more detailed analysis of the development of the wind profile with distance to the upwind area is only possible for the PALM simulation as there

are no further measurements available from the wind-tunnel experiment. We analyzed the development of the profile of the u component at varying distances to the upwind area at y=1400m which is close to station 3. Profiles are depicted within the annotated figure. The building area starts at x=3500m where the surface was raised from z=-5m (lowest height within the simulation) to z=0m which is in accordance to the wind-tunnel experiment. Profiles are shown until x=3800m which corresponds to the x coordinate of station 3. We assume that the PALM profiles match to the wind-tunnel profiles at x=3800m as measurements show at station 3. The depicted profiles do not change significantly from x=3600m onward. Hence, we conclude that the wind flow between both experiments should also not vary anymore starting from this point further down-wind. The differences of the approaching flow are therefore limited to the first about 100m of the building area (i.e. from x=3500m to x=3600m) at least close to the ground surface. Figure 2 also shows that the differences between both experiments are only small within the lower 100m and become larger at levels above 100m height. Further analysis of the wind field at heights above 100m were, however, not possible due to the lack of reference data at these heights. It is however likely that differences between PALM and wind-tunnel results would have been larger closer to the upwind area than to the downwind area.

**Answers to the specific comments**

- L7: Indeed we did not present any proof for our recommendations in how to reduce differences between measurements and simulation results. However, we discuss how differences might be reduced. We therefore rephrase the sentence: "In the end, we discuss how these differences might be reduced using already implemented features of PALM."

- L27: See answer to general comments.

- L71: Wind tunnel experiments were carried out long before corresponding numerical simulations took place. The choice of measurement locations in the wind tunnel was based on considerations regarding area coverage, location of observation stations as well as the investigated effect of building structure on near ground ventilation and pedestrian wind comfort. They were not specifically chosen with respect to average values of measurements covering the entire area of interest because this would have been too time consuming regarding the laboratory experiments. This approach was decided to be appropriate because the basic flow dynamics core in PALM had been evaluated successfully before already.

- L76: Crucial differences between PALM version 4.0 and version 6.0 are now mentioned within Sect. 1 of the updated manuscript. See also our answer to the general comments.

- L96: We agree that the wind speed of 1.54m/s is a rather weak wind speed which originated from a former misinterpretation of wind-tunnel results by the simulation team of our group. However, we think that our results still lie within an area of high Reynolds numbers and are therefore mostly independent from the actual wind speed value.

- L158, L167: The reference height was defined by qualified laboratory experiments to be representative for the measured canopy flow and is expected to be well within the height range for which a scaled neutrally stratified atmospheric boundary layer wind flow could be modeled most accurately. Despite the fact, that the reference height could have been changed when comparing numerical results with wind tunnel data, we decided not to introduce additional uncertainty in data comparison by translating reference conditions to a different height range based on common assumptions on the vertical structure of the ABL. For a neutrally stratified and fully turbulent near-ground ABL flow, the flow dynamics and

statistics of turbulence perfectly scales with the chosen reference wind speed and non-dimensional flow and turbulence data do not change within the bounds of experimental uncertainty. As a part of quality assurance of experimental data, this behavior is mandatory to be checked and documented before systematic experiments are carried out in a proper boundary layer wind tunnel experiment.

- L170: We tried to not add another layer of uncertainty to the results by interpolating them to different heights. Hence, we compared the non-interpolated values and accepted a small mismatch in measurement heights. Interpolation of results at greater heights might have been possible, however, closer to the ground, the uncertainty introduced by interpolation might have been as high as not interpolating the results at all. Therefore, we decided to not interpolate the results to different heights.

- L240: The local wind speed differs between PALM results and the wind-tunnel measurements at rooftop height at both stations as shown in Figs. 11 and 12. However, at higher levels, wind speeds agree between both experiments. Hence, we concluded that the mismatch is caused by differences within the building layout and surface roughness. This will then also result in a mismatch of local horizontal wind speed. As in both experiments the building heights are exactly the same in the close vicinity of stations 4 and 11, the differences must occur from the brick-like representation resulting in smooth walls being significantly more rough by the introduction of additional corners. Additionally, the mentioned mismatch of z0 between both experiments, where in PALM z0 was of higher value than in the wind-tunnel experiment, also leads to higher surface roughness within the PALM simulation and results in higher turbulence intensity and a shift of the turbulence spectra to higher wave numbers. As the overall wind speed is about similar within both experiments, the local wind speed is then reduced due to the higher roughness which can be observed in Figs. 11 and 12.

**Fig. 1.** Time averaged vertical profiles of the normalized u component from the PALM simulation at several x coordinates at y=1400m.

---

## Author Comment (AC2) · 15 Oct 2020

Within the published preprint, there are some minor mistakes we would like to correct:

- L98: a roughness length of 0.1m was used, not 1m.

- Fig.2: wrong height values were used for the plot of the approaching flow. This was an error purely within the plotting routine and had no further influence on the results. See attached the corrected figure (Fig. 1). The corrected figure caption will be: "Mean profiles of the approaching flow for the wind-tunnel experiment and PALM simulation. Note that z=0m is defined at street level height while the lowest level (white area) within

both experiments was at z=-5m."

- Fig.4: building heights were given relative to the bottom boundary of the simulation. However, within the manuscript, the street-level height is used as the reference height, i.e. z=0m. Hence, we changed the height labels within Fig. 4 in order to always refer to the same height levels throughout the entire text. See attached the corrected figure (Fig. 2). The corrected figure caption will be: "Building layout and heights as used in the PALM simulation. The x direction is oriented to follow the mean wind direction. The total domain size is 6000m, 2880m, and 601m in x, y and z direction, respectively. Note that z=0m is defined at street level height while the lowest level (white area) within both experiments was at z=-5m."

[Figure]

**Fig. 1.** Mean profiles of the approaching flow for the wind-tunnel experiment and PALM simulation.

**Fig. 2.** Building layout and heights as used in the PALM simulation.

---

## Referee Comment (RC2) · Anonymous Referee #2 · 19 Dec 2020

Review of

**Validation of the Dynamic Core of the PALM Model System 6.0 in Urban Environments: LES and Wind-tunnel Experiments**

By Gronemeier et al

The manuscript compares a modelled urban flow (obtained with a new PALM model version) to wind-tunnel measurements (representing a part of the city of Hamburg). The results are solid, but I am not sure if the employed double-blind strategy helps more than it distracts from the ultimate goal of validating the code and demonstrate the model capabilities. Moreover, I am not sure if "validation of the dynamic core" in the manuscript title oversells the results as only one case study with a single simulation is performed.

**Major comments**

1. In the beginning, I interpreted the title of the manuscript in a way that the dynamic core of the new PALM model version is validated. But I do think this is not achieved by limiting the validation to a single case study. The first sentence of the abstract reflects much better what was achieved here. Please adapt the title accordingly or expand the study to a comprehensive dynamical core validation.

2. Generally, I appreciate the double-blind test strategy. Nevertheless, at some point it would have been nice to see if the model-observation agreement is better when other values of the discussed parameters would have been chosen. In the present evaluation, any disagreement can be explained by a "wrong" setting of some parameter. In the end, the reader does not know, if the model could reproduce the observation if it had used better values of "external" parameters. In the end, it may turn into a philosophical question, how much a priori knowledge you allow to be included in your model setup? When I first read the manuscript, I thought "double-blind" is the optimal approach as it guarantees an unbiased and honest comparison. After thinking more about it, I felt that the double-blind test strategy leaves me unsatisfied as you often blame initial inconsistencies for the observed discrepancies. But what conclusions regarding the validation of the model remain at the end of the day (Q1)? In my impression, the manuscript gives more an answer to the question, how well can I simulate a specific scenario and what uncertainties remain having in mind that I do not know precisely which values to choose for some parameters in the beginning (Q2). Imagine the situation: You are asked to perform a PALM simulation for a different city where you obtain topography data but no measurement stations are installed for comparison. Then the present study helps assessing the faithfulness of your simulation results. It may help, if the manuscript makes clearer what part of the analysis relates more to answering question Q1 or Q2.
For example, lines 190-194: For the purpose of validation, wouldn't it be better to correct the topography data, eliminate this flaw and redo the simulation? In the present state, this flaw distracts more than it helps, at least when you pursue a model validation.
Moreover, I would have appreciated to see the sensitivity to the grid resolution (which I suppose can be varied independently of the resolution of the topography data).
The paper is not long, one could also think of adding the nesting feature to the present manuscript.

3. It is only marginally explained how the scaling works between the wind-tunnel scale and the real-world scale. In the simulation framework, I believe, it would have easily been possible to simulate the flow at the wind-tunnel scale? Why did you choose to simulate it on the real-world scale even though this can introduce ambiguities? Scaling relations for all displayed or discussed quantities should be derived, e.g. in a separate section in the appendix or in a table if it can be done in a compact way.
4. In general, my impression is that the comparison in Section 3.2 is not very elaborate and discusses only few properties. At least, the description of the results is rather subjective and uses often simple phrases.

**Minor comments**

In general, the figure captions sometimes lack crucial information to understand what it is shown or to make it easier for the reader.

1. I suppose the approaching flow displayed in Fig. 2 is part of the setup and not of the validation. Isn't it possible to prescribe in the model a flow that is closer to measured profile?
2. Figure 5: Which value range does the white colour represent? If the street level is at z=5m, the background colour should be blueish, shouldn't it.
3. Along which direction do streak-like structures appear? Why does it help to shift the flow in cross-stream direction? How does this prevent the formation of streak-like structures?
4. Line 138: I do not understand what you want to describe. "Profiles of nine different quantities…"(?) "The profiles were recorded with a time resolution of 2Hz, i.e. 9 evaluation during one model time step." (?)
5. Issue with staggered grid: Why not additionally evaluate the model flow also at z=3.5m and 10.5m? What's the wind speed deficit?
6. Fig. 8 and 9: You have to explain the meaning of the arrows (in particular, size and why you show differently sized arrows at each station).
7. The caption of Fig. 6 should tell me that the plot uses to different scales on the horizontal axis. Furthermore, you may replace the blue colour by some colour that can be more easily distinguished from the black line.

**Technical comments**

1. Abstract, line 1: "We **demonstrate"**
2. Line 2: I believe scenario fits better than situation: "The studied **scenario"**
3. In English language there is a difference between "which" and "that" many are not aware of. https://www.diffen.com/difference/That_vs_Which. Please go trough the whole manuscript. Line 35 is one such example.
4. Line 69: length -> period
5. Line 76: ". A detailed description" -> ", which"
6. Line 82: I guess the model DOMAIN was rotated?
7. Line 106: POTENTIAL temperature
8. Line 143: move the variable name in front of the given values (e.g. E=0.8865).
9. Line 153: "magnitude of rotation": Is this the vorticity magnitude?
10. Line 157: remove "to"
11. The paragraph from line 203 on should start with information that holds for all three Figures 10-12. In the next paragraph you should turn the attention the specific results of station 7.
12. Please reformulate line 228.

---

## Author Response (AR1)

**Author's response to the reviews**

At first we would like to express our gratitude for the valuable comments received from the referees. They precisely displayed the shortcomings of our study and helped us enormously to improve the quality of our manuscript and the overall study. After considering all comments and critics about our manuscript, we decided to re-work large parts of our study and remove all flaws within our experiment design following the comments and suggestions made by the referees.

The first referee noted that we need to be more specific about the changes between the old and the new PALM versions and why we need an updated evaluation of the code. This is now addressed in the revised manuscript. Also the miss-match of the approaching flow between both experiments raised concerns. This was related to an incorrect representation of the upwind area within the PALM simulation as was also discussed in the previous version of the manuscript. This miss-match, however, had no significant impact on the results of the comparison as discussed in the following. During the review process, it turned out, however, that we had to revise our simulation setup at several points and redo the simulation. Within the updated simulation setup, we then also revised the approaching flow to agree better with the wind-tunnel experiment.

The main comments made by the second referee were related to the many flaws in our simulation setup which we described and discussed within our manuscript. Even though all these spotted differences could be explained, it would have been easy to remove the respective errors from the setup, as was correctly pointed out by the referee. Hence, we decided, as recommended, to redo our simulations with an improved setup to better reproduce the wind-tunnel results. As expected, this improved the comparison at many places, while the general outcome of the study did not change much. Also, the depth of the analysis part of the manuscript was revised as suggested by the referee. This resulted in a much more elaborate comparison drawing a much better picture of the capabilities of PALM to simulate a realistic urban flow.

A detailed answer to all comments is listed below. Referee comments are written in *blue italic font*.

**Answers to the comments of referee 1**

**Answers to the general comments**

*This study has examined the representation of the turbulent flow characteristics reproduced by a newly updated CFD model, PALM6.0. The simulation results are compared with a wind tunnel experiment regarding the flow around a realistic building geometry in Hafencity. The results were generally well correspondence with the wind tunnel experiment, not only the simple mean wind speed but also the variance of the wind velocity and the spectral variation. This report will be useful for the users of this model. In addition, it also indicates some general issues in the simulation of the urban airflow. Therefore, this work is worth for publication. Besides, I still have some comments to be clarified before publication. One is about the model description. This study is motivated to analyze the model performance which is updated to version 6 of PALM. However, it is not sure if the core part of the model, which is related to the prognostic equation of the momentum (e.g. advection, diffusion, time-integration or wall boundary conditions, etc.) except the thermal effect, are updated from the previous version. Please describe more details about this point clearly. The results should also be more focused on the influence of the updated parts. Another point is the upwind condition. I think the results could be different near the upwind region if the inflow conditions are so different (Fig.2). However, the authors indicated that the observation points near the upwind region (e.g. station 2, 3, etc.) are almost same both in CFD and wind tunnel experiment. I am not sure why they are the same results irrespective of the different approach flows. Although it is fair to use the periodic boundary conditions in the streamwise direction, the effect of the inflow condition has to be more carefully examined (e.g. how far the direct influence of the inflow is observed?).*

The changes between PALM4.0 and PALM6.0 (version 5 does not exist) are manifold. A detailed description to the current state and the changes between both versions are given by Maronga et al. (2020, https://doi.org/10.5194/gmd-13-1335-2020). However, we agree that the reader should at least get an overview on the changes regarding the dynamic core of the model. The most significant changes within the code are more related to code structure and less to the physics. With the introduction of the land-surface and building surface models, the internal treatment of surfaces changed significantly. Several one-dimensional Fortran data structures were introduced to handle surface treatment. This also includes treatment of surfaces were neither the land-surface model nor the building-surface model is used as it is the case in our study. Also, further modularization of the code, i.e. reordering code parts like boundary treatment or turbulence closure into separate modules changed the overall code structure of the PALM model also affecting the model core. These changes should ideally have no effect on the simulation results but in reality might have small changes due to changes in computational order and rounding errors or even have changes due to fixed bugs which might have been not even noticed. Further changes which might have an effect on the results are the optimization of the multigrid solver which is also described by Maronga et al. (2020, https://doi.org/10.5194/gmd-13-1335-2020) and the calculation of the constant-flux layer which uses the Monin-Obukhov similarity theory (MOST) which was also used in previous versions of PALM but was also completely reworked for version 6.0.

[Figure]

Figure 1: Time averaged vertical profiles of the normalized $u$ component from the PALM simulation at several $x$ coordinates at $y = 1400\,\mathrm{m}$.

New code features, which were introduced in version 6.0 and which were used within this study, include the y-shift method and the virtual measurements module both of which we mentioned within our manuscript. We modified the introduction part of the manuscript in order to mention the changes between the old and the new PALM version.

The second part of the general commend made was regarding the inflow condition. It is mentioned that although the approaching flow shows clear differences between PALM and the wind-tunnel data, profiles are said to match well for stations 2 and 3 which are positioned close to the windward edge of the analysis area. Differences between the approaching flow are due to the different setup of the upwind area between both experiments. While roughness elements are present within the wind-tunnel experiment, there was only a flat surface present within the PALM simulation as discussed in Sect. 2.2 in the manuscript. A more detailed analysis of the development of the wind profile with distance to the upwind area is only possible for the PALM simulation as there are no further measurements available from the wind-tunnel experiment. We analyzed the development of the profile of the u component at varying distances to the upwind area at y=1400m which is close to station 3. Profiles are depicted within Fig. 1. The building area starts at $x = 3500\,\mathrm{m}$ where the surface was raised from $z = -5\,\mathrm{m}$ (lowest height within the simulation) to $z = 0\,\mathrm{m}$ which is in accordance to the wind-tunnel experiment. Profiles are shown until $x = 3800\,\mathrm{m}$ which corresponds to the $x$ coordinate of station 3. We assume that the PALM profiles match to the wind-tunnel profiles at $x = 3800\,\mathrm{m}$ as measurements show at station 3. The depicted profiles do not change significantly from $x = 3600\,\mathrm{m}$ onward. Hence, we conclude that the wind flow between both experiments should also not vary anymore starting from this point further downwind. The differences of the approaching flow are therefore limited to the first about 100 m of the building area (i.e. from $x = 3500\,\mathrm{m}$ to $x = 3600\,\mathrm{m}$) at least close to the ground surface.

However, after considering all review comments, we revised our simulation setup and improved the representation of the upwind area by including roughness elements of the same shape and layout as in the wind-tunnel experiment. This resulted in a significant improvement of the approaching flow where both experiments now show nearly identical inflow profiles.

**Answers to the specific comments**

- *L7: "In the end,..." This could not be discussed from the materials shown in the present manuscript.*
  Indeed we did not present any proof for our recommendations in how to reduce differences between measurements and simulation results. However, we discuss how differences might be reduced. We therefore rephrase the sentence: "In the end, we discuss how these differences might be reduced using already implemented

features of PALM."

- *L27: "drastic change..." I could understand there are many updates for the application parts but still not sure for the core prognostic parts. This is related to the general comment.*
  See answer to general comments.

- *L71: "25 locations..." How are these station chosen? For example, how the average of 25 vertical profiles becomes compared with the total horizontal average in the numerical simulation?*
  Wind tunnel experiments were carried out long before corresponding numerical simulations took place. The choice of measurement locations in the wind tunnel was based on considerations regarding area coverage, location of observation stations as well as the investigated effect of building structure on near ground ventilation and pedestrian wind comfort. They were not specifically chosen with respect to average values of measurements covering the entire area of interest because this would have been too time consuming regarding the laboratory experiments. This approach was decided to be appropriate because the basic flow dynamics core in PALM had been evaluated successfully before already.

- *L76: " we skip..." It has to be indicated here about which parts of the prognostic calculation were updated.*
  Crucial differences between PALM version 4.0 and version 6.0 are now mentioned within Sect. 1 of the updated manuscript. See also our answer to the general comments.

- *L96: "1.54 m/s..." This is rather weak at the top of the boundary layer if this is in the real atmospheric scale. Are the present results really free from this main wind speed?*
  We agree that the wind speed of $1.54\,\mathrm{m\,s^{-1}}$ is a rather weak wind speed which originated from a former misinterpretation of wind-tunnel results by the simulation team of our group. Within our revised simulation setup we addressed this issue and scaled the wind-speed profile to a value of $4\,\mathrm{m\,s^{-1}}$ at the reference height ($z = 50\,\mathrm{m}$). This resulted in a wind speed of $6.26\,\mathrm{m\,s^{-1}}$.

- *L158, L167,...: "the approaching flow at a height of 50 m" "wind speed is $23\,\%$ less..." Please explain why the wind speed at this height is the reference. Since the correspondence of the magnitude of the vertical profile directly depends on this parameter, it needs justification to use this parameter as a representative velocity scale.*
  The reference height was defined by qualified laboratory experiments to be representative for the measured canopy flow and is expected to be well within the height range for which a scaled neutrally stratified atmospheric boundary layer wind flow could be modeled most accurately. Despite the fact, that the reference height could have been changed when comparing numerical results with wind tunnel data, we decided not to introduce additional uncertainty in data comparison by translating reference conditions to a different height range based on common assumptions on the vertical structure of the ABL. For a neutrally stratified and fully turbulent near-ground ABL flow, the flow dynamics and statistics of turbulence perfectly scales with the chosen reference wind speed and non-dimensional flow and turbulence data do not change within the bounds of experimental uncertainty. As a part of quality assurance of experimental data, this behavior is mandatory to be checked and documented before systematic experiments are carried out in a proper boundary layer wind tunnel experiment.

- *L170: "located 0.5m lower" Why the values are not interpolated (e.g. linear interpolation, cubic spline, etc.) for the comparison?*
  We tried to not add another layer of uncertainty to the results by interpolating them to different heights. Hence, we compared the non-interpolated values and accepted a small mismatch in measurement heights. Interpolation of results at greater heights might have been possible, however, closer to the ground, the uncertainty introduced by interpolation might have been as high as not interpolating the results at all. Therefore, we decided to not interpolate the results to different heights.

- *L240: "At roof top height" How is the effect of the mismatch of the local horizontal wind speed in the numerical simulation and the wind tunnel? I think the ratio of the building sizes, which will be related to the peak wavelength at the roof level, and that of the local wind speed will be different for the numerical simulation and the wind tunnel experiment.*
  The local wind speed differs between PALM results and the wind-tunnel measurements at rooftop height at both stations as shown in Figs. 11 and 12. However, at higher levels, wind speeds agree between both experiments. Hence, we concluded that the mismatch is caused by differences within the building layout and surface roughness. This will then also result in a mismatch of local horizontal wind speed. As in both experiments the building heights are exactly the same in the close vicinity of stations 4 and 11, the differences must occur from the brick-like representation resulting in smooth walls being significantly more rough by the introduction of additional corners. Additionally, the mentioned mismatch of $z_0$ between both experiments, where in PALM $z_0$ was of higher value than in the wind-tunnel experiment, also leads to higher surface roughness within the PALM simulation and results in higher turbulence intensity and a shift of

the turbulence spectra to higher wave numbers. As the overall wind speed is about similar within both experiments, the local wind speed is then reduced due to the higher roughness which can be observed in Figs. 11 and 12.

**Answers to the comments of referee 2**

**Answers to the major comments**

**Comment 1**

*In the beginning, I interpreted the title of the manuscript in a way that the dynamic core of the new PALM model version is validated. But I do think this is not achieved by limiting the validation to a single case study. The first sentence of the abstract reflects much better what was achieved here. Please adapt the title accordingly or expand the study to a comprehensive dynamical core validation.*

The former version of the manuscript was lacking a proper evaluation of the results and only provided a basic comparison between wind-tunnel and PALM experiment. However, during the cause of the review process and thanks to the given comments by the referees, we were able to extend our study and present a proper evaluation of PALM in a neutrally stratified urban environment. Anyhow, the study still covers only a single case study without showing the general performance for different cases. Therefore, a complete validation study is still not achieved. Hence, we adjusted the title: *"Evaluation of the Dynamic Core of the PALM Model System 6.0 in a Neutrally Stratified Urban Environment: Comparison between LES and Wind-tunnel Experiments"*.

**Comment 2**

*Generally, I appreciate the double-blind test strategy. Nevertheless, at some point it would have been nice to see if the model-observation agreement is better when other values of the discussed parameters would have been chosen. In the present evaluation, any disagreement can be explained by a "wrong" setting of some parameter. In the end, the reader does not know, if the model could reproduce the observation if it had used better values of "external" parameters. In the end, it may turn into a philosophical question, how much a priori knowledge you allow to be included in your model setup? When I first read the manuscript, I thought "double-blind" is the optimal approach as it guarantees an unbiased and honest comparison. After thinking more about it, I felt that the double-blind test strategy leaves me unsatisfied as you often blame initial inconsistencies for the observed discrepancies. But what conclusions regarding the validation of the model remain at the end of the day (Q1)? In my impression, the manuscript gives more an answer to the question, how well can I simulate a specific scenario and what uncertainties remain having in mind that I do not know precisely which values to choose for some parameters in the beginning (Q2). Imagine the situation: You are asked to perform a PALM simulation for a different city where you obtain topography data but no measurement stations are installed for comparison. Then the present study helps assessing the faithfulness of your simulation results. It may help, if the manuscript makes clearer what part of the analysis relates more to answering question Q1 or Q2.*

*For example, lines 190-194: For the purpose of validation, wouldn't it be better to correct the topography data, eliminate this flaw and redo the simulation? In the present state, this flaw distracts more than it helps, at least when you pursue a model validation. Moreover, I would have appreciated to see the sensitivity to the grid resolution (which I suppose can be varied independently of the resolution of the topography data). The paper is not long, one could also think of adding the nesting feature to the present manuscript.*

We agree that the comparison suffers from the many discovered incorrect settings used for the simulation and re-simulating the case with a revised setup would improve the reliability of the study. Therefore, after thoroughly assessing our available resources, we decided to re-simulate the study. We however did not consider the nesting feature as this would have required further tests and introduce another source of possible deviations between PALM and wind-tunnel results. Unfortunately, our available computing resources did not allow us to perform a sophisticated grid-sensitivity study. Hence we were only able to redo the main simulation.

The following parts of the simulation setup were corrected:

- Building height near measurement station 13 was corrected from 24 m to 6 m.

- Roughness elements of same shape and layout as in the wind-tunnel experiment were placed around the building setup.

- $z_0$ was reduced following the recommendation given by Basu and Lacser (2017, https://doi.org/10.1007/s10546-016-0225-y) who state that $z_0$ should be no larger than 0.02 times the first grid-level height. This resulted in $z_0 = 0.01$ m in our case.

- During the review process, another referee pointed out that the wind speed of $1.54 \text{m s}^{-1}$ at the top of the simulation domain is rather weak (see answers to referee 1). As we pointed out in the answers to the referee's comment, this wind speed resulted from a misinterpretation of the wind-tunnel measurements by the

simulation team of our group. Even though we do not think that this affects the results of the comparison, we also addressed this in the revised simulation setup and increased the approaching wind-speed profile by a factor of 4 at all heights.

All these changes between the former and the revised simulation only affect the setup and could have been done even without the knowledge of the actual wind-tunnel measurements. Especially, we did not tune our model physics in order to increase the agreement between simulation and wind-tunnel measurements. However, due to the fact that some of the setup mistakes were only discovered after comparing PALM and wind-tunnel measurements, our study does not fulfill the requirements of a blind study any longer. The experience we made from this is, to very carefully prepare and check the simulation setup. Especially matching the roughness to the wind-tunnel experiment is of high importance as this has a strong influence on the near-surface results and the approaching wind profile.

Due to the increased roughness introduced by the roughness elements, the approaching flow of the PALM simulation now agrees well to that of the wind-tunnel experiment. Results at station 13 now also agree between both experiments to the same degree as those of the other measurement stations. The reduced $z_0$ resulted in an overall better agreement between PALM and wind-tunnel measurements at the lowest evaluated heights. However, the general outcome of the study did not change by the revised simulation setup. In particular, the discussion about deviations between PALM and wind-tunnel results at station 11 and 7 is still valid.

Section 3 of the manuscript was entirely reworked to account for the changes of the simulation setup and the more detailed evaluation as also mentioned in the answer to comment 4.

**Comment 3**

*It is only marginally explained how the scaling works between the wind-tunnel scale and the real-world scale. In the simulation framework, I believe, it would have easily been possible to simulate the flow at the wind-tunnel scale? Why did you choose to simulate it on the real-world scale even though this can introduce ambiguities? Scaling relations for all displayed or discussed quantities should be derived, e.g. in a separate section in the appendix or in a table if it can be done in a compact way.*

We decided to operate PALM in full scale (real-world scale) because this is closer to a realistic use-case for a PALM simulation than running PALM in wind-tunnel scale. Although this requires to scale up the results from the wind-tunnel experiment to full scale, we still think this way the results can be of better use to other studies also running PALM at full scale because there is no scale difference between our validation study and future studies applying PALM at full scale. The scaling of the wind-tunnel results is done using the scaling factor $m = l_{\mathrm{ms}}/l_{\mathrm{fs}} = 1/500$ of the model, where the subscripts "ms" and "fs" denote "model scale" and "full scale", respectively, and $l$ denotes distance. From $t = u \cdot l$, where $t$ is time and $u$ wind speed, it follows:

$$t_{\mathrm{fs}} = t_{\mathrm{ms}} \frac{l_{\mathrm{fs}}}{l_{\mathrm{ms}}} \frac{u_{\mathrm{ms}}}{u_{\mathrm{fs}}}.$$

Further, it is assumed that $u_{\mathrm{ms}} = u_{\mathrm{fs}}$. Hence, lengths are scaled via $l_{\mathrm{fs}} = \frac{l_{\mathrm{ms}}}{m}$ and times via $t_{\mathrm{fs}} = \frac{t_{\mathrm{ms}}}{m}$. We added this information to Sec. 2.1.

**Comment 4**

*In general, my impression is that the comparison in Section 3.2 is not very elaborate and discusses only few properties. At least, the description of the results is rather subjective and uses often simple phrases.*

In retrospect, we agree that the comparison was not as elaborate as it should be. We therefore increased the number of analyzed quantities to draw a more detailed picture of the performance of PALM. Additional quantities are the skewness, kurtosis and length scales of the wind velocity components and the variation of the wind direction. Further, we support our comparison results by showing scatter diagrams of various quantities for all measurement stations together with the vertical profiles at three example stations. To be more comparable to other studies and to increase objectiveness, we now calculated several validation metrics like the factor-of-two and the hit rate and report their values.

**Answers to the minor comments**

1. *I suppose the approaching flow displayed in Fig. 2 is part of the setup and not of the validation. Isn't it possible to prescribe in the model a flow that is closer to measured profile?*
   Yes, the inflow profile is not part of the evaluation but belongs to the setup. Differences within the inflow profile were mainly caused by the absence of roughness elements within the upwind area during the PALM simulation. Within the wind-tunnel experiment roughness elements were used to in increase surface roughness of the upwind area to $z_0 = 0.66$m. Within the PALM simulation, the upwind area was flat and the prescribed surface roughness was set to $0.1\,\mathrm{m}$. Due to the implemented consideration of the constant flux

layer within PALM and the chosen grid resolution of 1 m for the simulation, we could not use $z_0 = 0.66$ m within the simulation but had to reduce the roughness to $z_0 = 0.1$ m as discussed in the previous version of the manuscript. Antoniou et al. (2017, https://doi.org/10.1016/j.buildenv.2017.10.013) also reported this behaviour when comparing LES and wind-tunnel results. Tests with a coarser resolution of 10 m grid size showed, that when using $z_0 = 0.66$ m, the approaching flow profile becomes more similar to that of the wind-tunnel experiment.

However, within the revised simulation setup, we included roughness elements of same shape and layout as in the wind-tunnel experiment as we also suggested in our non-revised manuscript. As a result, the inflow profile remarkably improved and both experiments now show nearly identical inflow profiles which also proves our hypothesis that the roughness elements must be considered to get a better match to the wind-tunnel experiments.

2. *Figure 5: Which value range does the white colour represent? If the street level is at z=5m, the background colour should be blueish, shouldn't it.*

    Within the original Fig. 5 the white background colour represented a height level of 0m. The white colour was not, however, part of the given colour range. We changed the figure so that only colours from the colour bar are used.

3. *Along which direction do streak-like structures appear? Why does it help to shift the flow in cross-stream direction? How does this prevent the formation of streak-like structures?*

    Streak-like structures appear along the mean flow direction. These structures usually span over large distances along the mean wind direction and do not move along the span-wise direction. Because the length of the streaks along stream-wise direction is significantly larger than the model domain size and because we use cyclic boundary conditions, streaks of infinite size are created. These infinite streaks are very stable in time and space. To interrupt and force a natural dissipation of these streaks, we used the shifting method developed and tested by Munters et al. (2016, https://doi.org/10.1063/1.4941912). The shifting does not prevent the streaks to develop but it prevents the formation of unnaturally stable structures. The text is changed to be more precise on this topic.

4. *Line 138: I do not understand what you want to describe. "Profiles of nine different quantities…"(?) "The profiles were recorded with a time resolution of 2Hz, i.e. 9 evaluation during one model time step." (?)*

    At each measuring site, vertical profiles were recorded within the LES at the horizontal grid position closest to the measuring coordinates and also at the surrounding eight grid points. So, in total there were vertical profiles recorded at nine different horizontal positions for each measuring site. We re-formulated the text to be less confusing.

5. *Issue with staggered grid: Why not additionally evaluate the model flow also at z=3.5m and 10.5m? What's the wind speed deficit?*

    Due to the staggered Arakawa C grid used in PALM and the chosen vertical resolution of 1 m, there was no grid point at the exact same height within the PALM simulation compared to the wind-tunnel measurements. Measurement heights within the PALM simulation were located either 0.5 m below or above the wind-tunnel measurements. When using either the lower or the higher grid layer, the mean deviation between PALM and wind-tunnel results changes from -8.7% to -5.4% (-5.4% to -4.7%) for the 3 m (10 m) height level. Hence, the deviation differs by 3.3% (0.7%) depending on the chosen grid level. The majority of the deviation however comes from the difference in roughness of the building layer introduced by the step-like representation of the topography and (especially at the lowest height level) presumably an overestimated $z_0$ as already mentioned in the previously submitted manuscript. At greater height, there is no difference in comparing the grid point below or above to the wind-tunnel measurements as expected. Validation metrics also change by less than 0.03 if the grid levels below or above the wind-tunnel heights are chosen. Within the manuscript we now give more information on this issue. We, however, continue to use the grid levels below the wind-tunnel heights for the majority of the evaluation.

6. *Fig. 8 and 9: You have to explain the meaning of the arrows (in particular, size and why you show differently sized arrows at each station).*

    Figure 8 and 9 depict wind roses at each measuring site. We omitted to show the axes at each of the wind roses for better overview of the figures. However, we now added axes to a single wind rose as reference and give more information within the figure caption.

7. *The caption of Fig. 6 should tell me that the plot uses to different scales on the horizontal axis. Furthermore, you may replace the blue colour by some colour that can be more easily distinguished from the black line.*

    We separated the single diagram of Figure 6 into two separated diagrams for improved readability. We also updated the figure caption accordingly.

**Answers to the technical comments**

Due to the intensive rework of large parts of the manuscript, some of the technical comments were obsolete. Where still applicable, comments were considered and the manuscript was changed accordingly.

---

## Author Response (AR2)

**Author's response to the reviews**

We would like to thank the referee to again review our manuscript.

The comments, given by the referee, are:

> I appreciate the substantial revision of the manuscript. The more elaborate comparison and giving up the double-blind strategy makes the manuscript much better in my opinion.
>
> Concerning my comment 2 of the previous review round, the rationale of the scaling should be better explained. First of all, the Navier-Stokes equations are non-linear, and in theory the flow is not scale independent. How can you be sure that PALM of the full scale and operated on the wind tunnel scale give basically identical results. Please clarify this.
>
> There are many language mistakes, in particular, in the new text parts. It is certainly not the task of the reviewer to make up for this, in particular when I have the impression, that the mistakes could have been eliminated if all co-authors read the new manuscript version.

As mentioned in our answer during the previous review round, PALM's main purpose is to simulate the atmospheric flow at full scale. The majority of studies and applications operate PALM at full scale rather than at wind-tunnel scale. Therefore, we think that it is of more use for future studies to evaluate PALM at full scale. Hence, we had to scale up the wind-tunnel results to be able to use them as reference for our evaluation. The scaling of wind-tunnel results is based on Townsend's hypothesis of the self similarity of fully rough turbulent flows (Townsend, A. A. (1956): "The structure of turbulent shear flow", Cambridge University Press, Cambridge, UK). According to this hypothesis, a turbulent flow is self similar, and hence, scale independent as soon as a fully rough flow regime is reached. During the wind-tunnel experiment, measurements were made to ensure that a fully rough turbulent flow regime has been developed. This allowed us to scale-up the measurements as described in the manuscript and compare them to our simulation results at full scale. This is a well-established method and a standard procedure for wind-tunnel experiments. Therefore, we did not describe this procedure in detail in our manuscript. If PALM results should be scaled down from full scale to wind-tunnel scale, similar tests need to be carried out beforehand to prove scale-independence for the simulation results. This, however, was not done in our study and we did not consider scaling the PALM simulations to a different scale.

We thoroughly checked the manuscript for language-related issues and corrected all spotted mistakes to the best of our knowledge.